# Breathing-driven prefrontal oscillations regulate maintenance of conditioned-fear evoked freezing independently of initiation

Sophie Bagur [1,4✉], Julie M. Lefort [1,4], Marie M. Lacroix[1], Gaëtan de Lavilléon[1], Cyril Herry [2,3], Mathilde Chouvaeff[1], Clara Billand[1], Hélène Geoffroy[1] & Karim Benchenane [1✉]

Brain–body interactions are thought to be essential in emotions but their physiological basis remains poorly understood. In mice, regular 4 Hz breathing appears during freezing after cue-fear conditioning. Here we show that the olfactory bulb (OB) transmits this rhythm to the dorsomedial prefrontal cortex (dmPFC) where it organizes neural activity. Reduction of the respiratory-related 4 Hz oscillation, via bulbectomy or optogenetic perturbation of the OB, reduces freezing. Behavioural modelling shows that this is due to a specific reduction in freezing maintenance without impacting its initiation, thus dissociating these two phenomena. dmPFC LFP and firing patterns support the region's specific function in freezing maintenance. In particular, population analysis reveals that network activity tracks 4 Hz power dynamics during freezing and reaches a stable state at 4 Hz peak that lasts until freezing termination. These results provide a potential mechanism and a functional role for bodily feedback in emotions and therefore shed light on the historical James–Cannon debate.

---

[1] Team Memory, Oscillations and Brain States (MOBs), Brain Plasticity Unit, CNRS, ESPCI Paris, PSL University, Paris, France. [2] INSERM, Neurocentre Magendie, Bordeaux, France. [3] University of Bordeaux, Neurocentre Magendie, Bordeaux, France. [4] These authors contributed equally: Sophie Bagur, Julie M. Lefort. ✉email: sophie.bagur@espci.fr; karim.benchenane@espci.fr

  1

Breathing is one of the most fundamental rhythms generated by the brain. Given its primary function of supplying the body with oxygen, it is exquisitely regulated to reliably meet metabolic demands[1]. The breathing rhythm does not just play a role in pumping air in and out of lungs: like most other physiological functions, it is modulated by emotion[2] but breathing stands out in that it is also modulated by a wide range of behaviours, such as olfactory sampling or vocalization and is under voluntary control[2–7]. A further particularity is that this rhythm also generates oscillatory activity in a wide range of brain areas. Indeed, its impact on hippocampal activity has long been known[8] and recent studies have extended this observation to many other regions, in particular, the cortex and limbic system in rodents[8–17] and also in humans[18,19]. Numerous lines of evidence suggest that the respiratory rhythm is principally relayed to these structures via the olfactory bulb (OB)[9,12,16] although somatosensation could also be involved[14]. This rhythm is originally generated in the brainstem[20] which induces muscular movement and causes air to flow through the nose. This rhythmically activates olfactory epithelial neurons because of the influx of air charged with odorant molecules. This entrainment is partially independent of olfaction since mechanosensitive epithelial neurons are also entrained by airflow[21]. In turn, the numerous projections of the OB entrain a wide range of brain regions.

What is the function of this rhythm that is now gaining recognition as an ubiquitous organizer of dynamics across multiple brain areas? The first proposed function is that the rhythmic feedback from olfactory centers may coordinate multiple orofacial activities with olfaction, such as chewing, swallowing, licking, vocalizing and, in rodents, whisking[22]. However, the breathing rhythm is present in non-sensory structures, so it could also play a role in other aspects of brain function.

Recent results indeed suggest that the breathing rhythm is involved in emotional behaviour[23]. However, as we shall briefly review, these results do not yet form a coherent picture. In rodents, auditory cue fear conditioning is the most widely used paradigm to evoke a fear-like emotional state[24], most often quantified using the robust freezing behaviour. In the prefrontal cortex (PFC), particularly in the dorsomedial region (dmPFC), a strong 4 Hz oscillation has been identified during freezing[25,26]. This rhythm organizes neural activity within this structure and with the amygdala. Increasing this rhythm with optogenetics increases freezing behaviour whereas its perturbation reduces freezing levels[25,26]. A recent study identified the origin of this 4 Hz oscillation as being a change in breathing rhythm during freezing that is transmitted via the OB to the dmPFC[23]. However, pharmacological disruption of the OB and therefore of the 4 Hz oscillation leads to an increase in cue-induced freezing behaviour, the opposite effect of that found using direct manipulation of the dmPFC 4 Hz[25]. Current knowledge does not allow to reconcile these contradictory results, suggesting that the function and mechanism of the respiratory-related 4 Hz oscillation in fear behaviour remains to be understood.

Moreover, the exact role of the PFC in fear-related behaviour is still open to debate. Original studies showed that the PFC was involved in the extinction of fear behaviour[27]. More recently, this function has been restricted to the infralimbic area, whereas it has emerged that the dorsomedial and in particular the prelimbic area plays a role in freezing expression (refs. [28,29], see ref. [30] for review). Whether there is a common framework to understand the two functions and how they are to be reconciled with PFC function in general, is still unresolved[29–31]. In particular, the PFC is generally viewed as an integrative region providing top-down inhibitory control[32]. Fear extinction is a context-dependent top-down regulation of a previously learned behavioural response and therefore fits very well with this view of PFC function. However, a direct role for the dorsomedial subregion in piloting freezing behaviour, whose main effector regions are subcortical, appears hard to reconcile with the broader role of the PFC. Since the 4 Hz rhythm is important to dmPFC function during freezing, we use manipulation of the origin of this rhythm, the OB, to probe the function of this area. This approach is similar to the manipulation of the theta oscillation that originates outside the hippocampus in order to understand hippocampal function.

The main goal of the paper is to make a thorough investigation of the role of the 4 Hz originating in the OB and its impact on dmPFC activity and freezing behaviour. First, we combined electrophysiological recordings and behavioural manipulations to assess and compare the effects of pharmacological, optogenetic and surgical disruption of OB activity. This allowed us to clearly show that loss of 4 Hz is associated with decreased freezing levels. Second, to understand the exact role of dmPFC 4 Hz, we analysed the dynamical regulation of this behaviour. Using a Markov model of behaviour to differentiate freezing initiation and maintenance, we found that 4 Hz activity plays a key role in maintaining ongoing freezing, without any effect on initiation of this behaviour. Based on this framework, we show that 4 Hz modulated single units in the dmPFC fired most strongly during the sustained plateau of freezing and freezing offset, coherent with a role in controlling freezing maintenance. Taken together, these findings point to a new model of regulation of emotional states.

## Results

**During freezing, regular 4 Hz breathing entrains the OB.** In order to study the influence of breathing-related oscillations on fear expression, we conditioned mice using a classical auditory cue conditioning paradigm. In test sessions 24 h later, animals demonstrated robust freezing in response to presentation of the paired stimulus (CS+) but not the unpaired stimulus (CS−) (Fig. 1a). We recorded breathing using full-body plethysmography and found that during active exploration of the environment mice breathed irregularly at frequencies ranging from 3 to 12 Hz with an increase to 8–15 Hz when highly active (Fig. 1b, c). During freezing, the characteristics of breathing were deeply modified (Fig. 1b): the frequency dropped to a narrow frequency band around 4 Hz (Fig. 1c), the tidal volume increased relative to non-freezing periods (Fig. 1d) and the variability of both frequency and tidal volume dropped (Fig. 1e, f), all in agreement with the previous observations[23].

The OB is entrained by the rhythmic airflow induced by breathing[21,33]. Accordingly, we found that OB LFP was coherent with respiration and in particular during freezing, it showed a prominent 4 Hz oscillation (Fig. 1b, g, h). Therefore, during freezing, breathing shifts to a slower, deeper and more regular 4 Hz rhythm which creates a 4 Hz respiratory-related rhythm in the OB, consistent with the previous reports[23,34].

**During freezing, dmPFC local activity is entrained by 4 Hz OB respiratory-related rhythm.** The OB provides a pathway via which the respiratory rhythm induces neural oscillatory activity in downstream brain regions[9,12,16]. We recorded LFP and single unit activity in the dorsomedial PFC (Fig. S1), an important region for freezing-related behaviour. dmPFC LFP displayed a strong 4 Hz oscillation during freezing that tracked OB activity (Figs. 1b, 2a) and was highly coherent with it (Fig. 2a, b, e, f, h), in agreement with previous studies[23,25,26]. Moreover, Granger causality shows that the OB clearly drives the dmPFC (Fig. 2c) in the 4 Hz band. Finally, both total and partial bulbectomy abolished dmPFC 4 Hz oscillations (Fig. 2d, g, Fig. S2H), similar to previously published results using naris occlusion or destruction

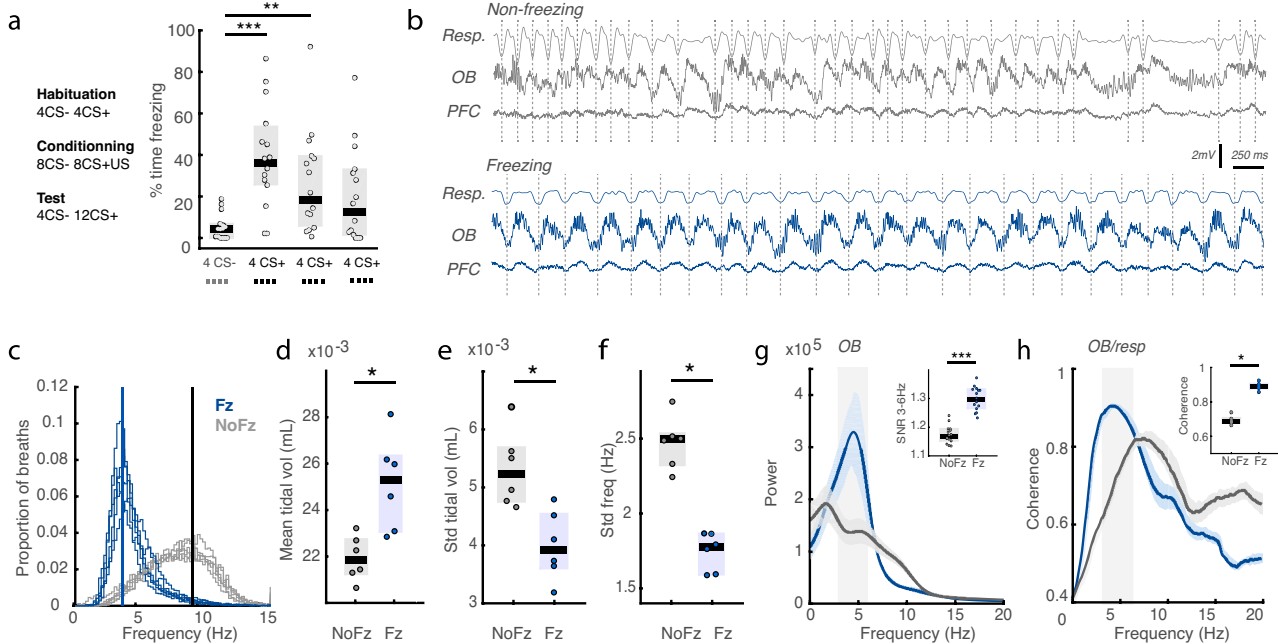

**Fig. 1 During freezing, regular 4 Hz breathing entrains the olfactory bulb. a** Experimental protocol and behavioural results: median percentage of time spent freezing for all mice during the CS− and CS+, grouped as blocks of four sounds (4CS− followed by 12 CS+). Sound block significantly affected freezing rates (Friedman test: $X^2(3) = 19.6$, $p = 2.01e−4$). Post-hoc analysis revealed significant differences of the CS+ blocks compared to the CS− block (Wilcoxon signed-rank test: Signed Rank statistic = 0,14,23; $p = 6e−5$, 0.0067, 0.0353, $n = 15$). **b** Illustrative example of breathing activity as recorded in the plethysmograph and OB and dmPFC LFPs during exploratory and freezing periods. Note the striking regularity and synchronicity in the 3–6 Hz range in all three time courses during freezing. **c** Distribution of breathing frequencies recorded in the plethysmograph during freezing and non-freezing periods. ($n = 6$). **d** Mean breathing tidal volume during freezing and non-freezing periods. (Paired Wilcoxon signed-rank test: Signed Rank statistic = 0, $p = 0.0312$, $n = 6$). **e**, **f** Variability of breathing frequency and tidal volume during freezing and non-freezing periods. (Paired Wilcoxon signed-rank test: Signed Rank statistic = 0,0; $p = 0.0312$, 0.0312, $n = 6$). **g** Averaged OB LFP power spectra during freezing (blue) and non-freezing (grey) periods. Error bars are SEM. Inset: average signal-to-noise ratio of 3–6 Hz band during freezing and non-freezing periods. The signal to noise ratio is defined as the ratio between power in the band of interest to the power in the rest of the spectrum. (Paired Wilcoxon signed-rank test: Signed Rank statistic = 0, $p = 6.10e−4$, $n = 15$). **h** Averaged coherence between breathing and the OB during freezing (blue) and non-freezing (grey) periods. Error bars are SEM. Inset: average coherence of 3–6 Hz band during freezing and non-freezing periods. (Paired Wilcoxon signed-rank test: Signed Rank statistic = 0, $p = 0.0312$, $n = 6$). In all panels, boxplots show the median and interquartile.

of the olfactory epithelium[12,23]. This demonstrates that the 4 Hz dmPFC oscillation originates in the OB respiratory-related rhythm.

Activity of the OB generates very large electrical fields and is situated in close vicinity to the dmPFC, raising the possibility that the LFP recorded in the dmPFC could be volume-conducted. However, we found that the 4 Hz signal remained intact when using bipolar derivation of the signal in the dmPFC, a proxy for local activity (Fig. S3). Moreover, we found that artificially induced oscillations in the OB were not linearly transmitted to the dmPFC but instead transmission depended non-linearly on frequency (Fig. S4). This is incompatible with volume conduction that depends on the passive electrical properties of biological tissues. Indeed, when we rhythmically activated OB interneurons (ChR2 in GAD-Cre mice) using optogenetics at 4 Hz, 7 Hz, 10 Hz and 13 Hz, this yielded a response that decreased with frequency in the OB, as expected. However in the dmPFC, we found an almost complete suppression of 10Hz-transmitted activity relative to the neighbouring 7 Hz and 13 Hz frequencies (Fig. S4). Furthermore, another study using current source density analysis showed that OB input generated a sink in deep dmPFC layers[34]. Therefore at least part of the 4 Hz power in dmPFC LFP is genuinely due to local activity.

Besides, during freezing, 45% of dmPFC units recorded were phase-modulated by the OB 4 Hz oscillation (Fig. 2i, j, Fig. S1C, Fig. S5). Moreover, we demonstrated both the necessity and sufficiency of the OB for this entrainment. First, we lesioned the olfactory epithelium using methimazole (described in the next section) and found a reduction in the proportion of dmPFC units entrained by the respiratory rhythm after fear conditioning (Fig. S6). Second, we stimulated OB interneurons using optogenetics and this stimulation efficiently entrained over a third of dmPFC single units (Fig. S4G, H).

We found that single units in the dmPFC were entrained more strongly and in larger numbers by the respiratory rhythm during freezing (3–5 Hz) than during the active state (6–15 Hz) (Fig. 2i, k, l and Fig. S7 for detailed methodology). Importantly, this result cannot be attributed to modification in overall spike count because individual neurons' change in modulation depth between freezing and active periods did not correlate with their change in firing rate (Fig. S8C). Overall, this demonstrates not only that the OB oscillation entrains dmPFC local activity but that this modulation is at its most efficient during freezing.

**4 Hz rhythm is necessary for freezing expression.** To decipher the functional role of this 4 Hz respiratory rhythm transmitted via the OB to the dmPFC, we performed bilateral ablation of the OBs. We found that after fear conditioning, bulbectomized mice displayed lower levels of freezing than the sham controls during the test sessions (Fig. 3a), consistent with results from the rat[35]. It is important to note that bulbectomy is known to induce broad behavioural changes such as depression-like symptoms and open-field hyperactivity. However, these modifications are never

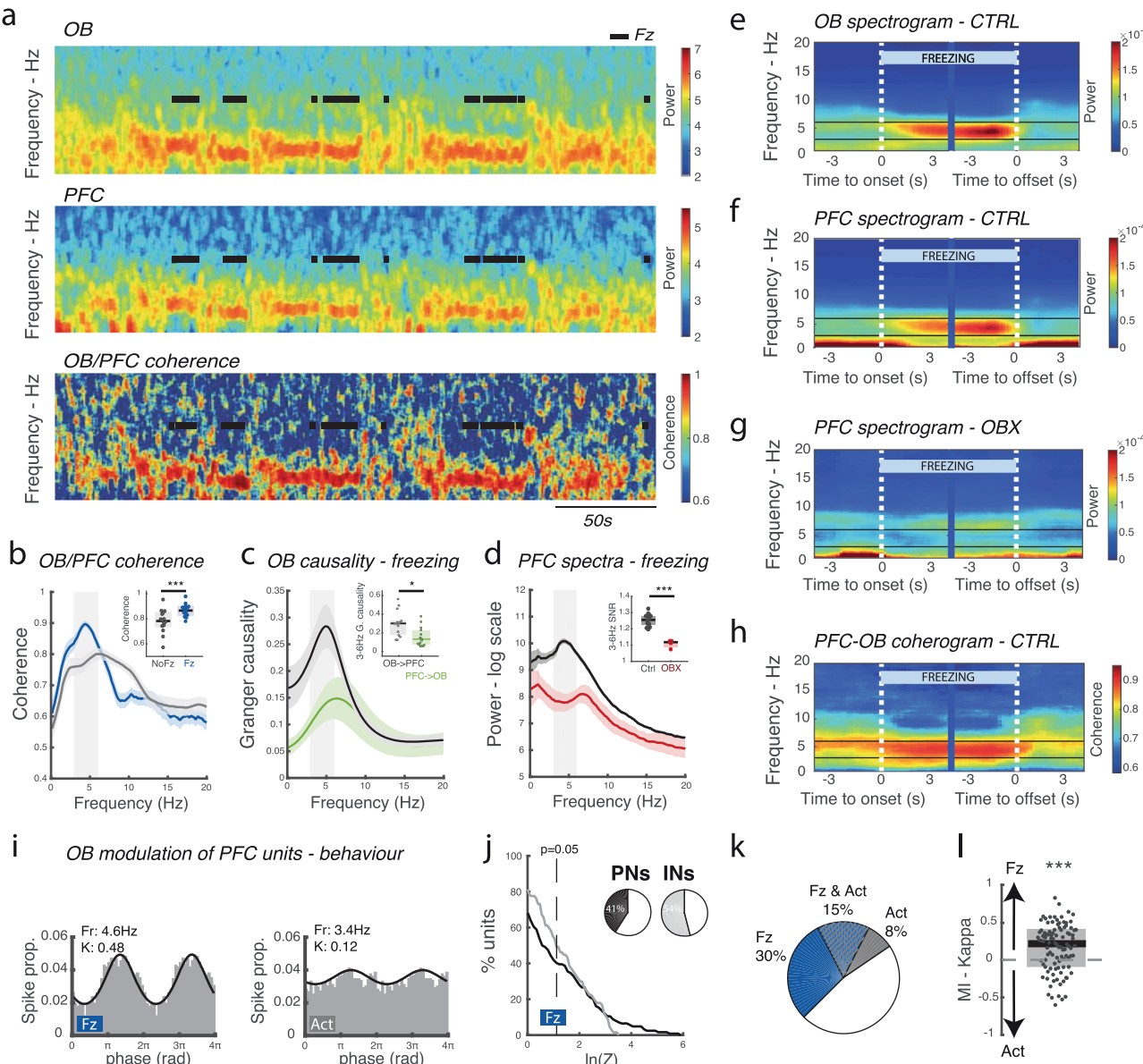

**Fig. 2 Breathing-related OB 4 Hz organizes neural firing in the dorsomedial prefrontal cortex during freezing. a** Representative spectrogram of OB and dmPFC activity and associated coherogram during a test session. Note the prominent 4 Hz oscillation in both structures during freezing epochs (black lines). **b** Averaged coherence between the OB and dmPFC during freezing (blue) and non-freezing (grey) periods. Error bars are SEM. Inset: average coherence of 3–6 Hz band. (Paired Wilcoxon signed-rank test: Signed Rank statistic = 1, $p = 6.1e-5$, $n = 15$). **c** Averaged Granger causality during freezing from OB to dmPFC (black) and from dmPFC to OB (green). Error bars are SEM. Inset: mean Granger causality in the 3–6 Hz band. (Paired Wilcoxon signed-rank test: Signed Rank statistic = 97, $p = 0.0353$, $n = 15$). **d** Averaged power spectra during freezing in sham (black) and bulbectomized (OBX) (red) mice. Error bars are SEM. Inset: signal-to-noise ratio of 3–6 Hz band during freezing periods. (Wilcoxon rank-sum test: Signed Rank statistic = 180, $p = 5.1e-4$, $n = 15$ and $n = 4$). **e** Averaged OB spectrogram triggered on freezing onset and offset in control mice. Note the appearance of strong activity in the 4 Hz band. **f**, **g** As in (**e**) for the dmPFC in control (**f**) and bulbectomized (**g**) mice. Note the appearance of 4 Hz in control but not in bulbectomized mice. **h** Averaged dmPFC-OB coherogram triggered on freezing onset and offset in control mice. **i** Phase histograms of an example dmPFC unit modulation by OB LFP showing clear gain in phase locking during freezing (left) relative to active (right) epochs. **j** Cumulative distribution of log-transformed Rayleigh's test Z of dmPFC putative principal neurons (PNs) and interneurons (INs) modulation by OB LFP. Inset: percentage of significantly modulated neurons using Rayleigh's test with $p = 0.05$. **k** Percentages of dmPFC units modulated by the OB LFP during active behaviour and during freezing using Rayleigh's test with $p = 0.05$. The proportion of OB modulated neurons increases significantly for freezing state. (45% vs 23%, chi2stat = 9.82, $p = 0.0017$, $n = 100$ units). **l** Modulation index of firing rate of all neurons between freezing and active periods. On average units decrease their firing rate during freezing. (One-sample two-sided Wilcoxon signed-rank test with 0: zval = −3.4314, $p = 6.0e-04$, $n = 100$ units). In all panels, boxplots show median and interquartile range.

observed in the 2-week window following bulbectomy. We thus performed all experiments before this time[36,37]. Moreover, to carefully control for effects of bulbectomy, we investigated the impact of both partial and total bulbectomy. Both yielded

reductions in freezing levels (Fig. 3a, Fig. S2I) with neither signs of locomotor changes in the test box before conditioning (Fig. S2J, K), nor open-field hyperactivity at the time of testing (Fig. S2B, C). Consistent with the literature, hyperactivity

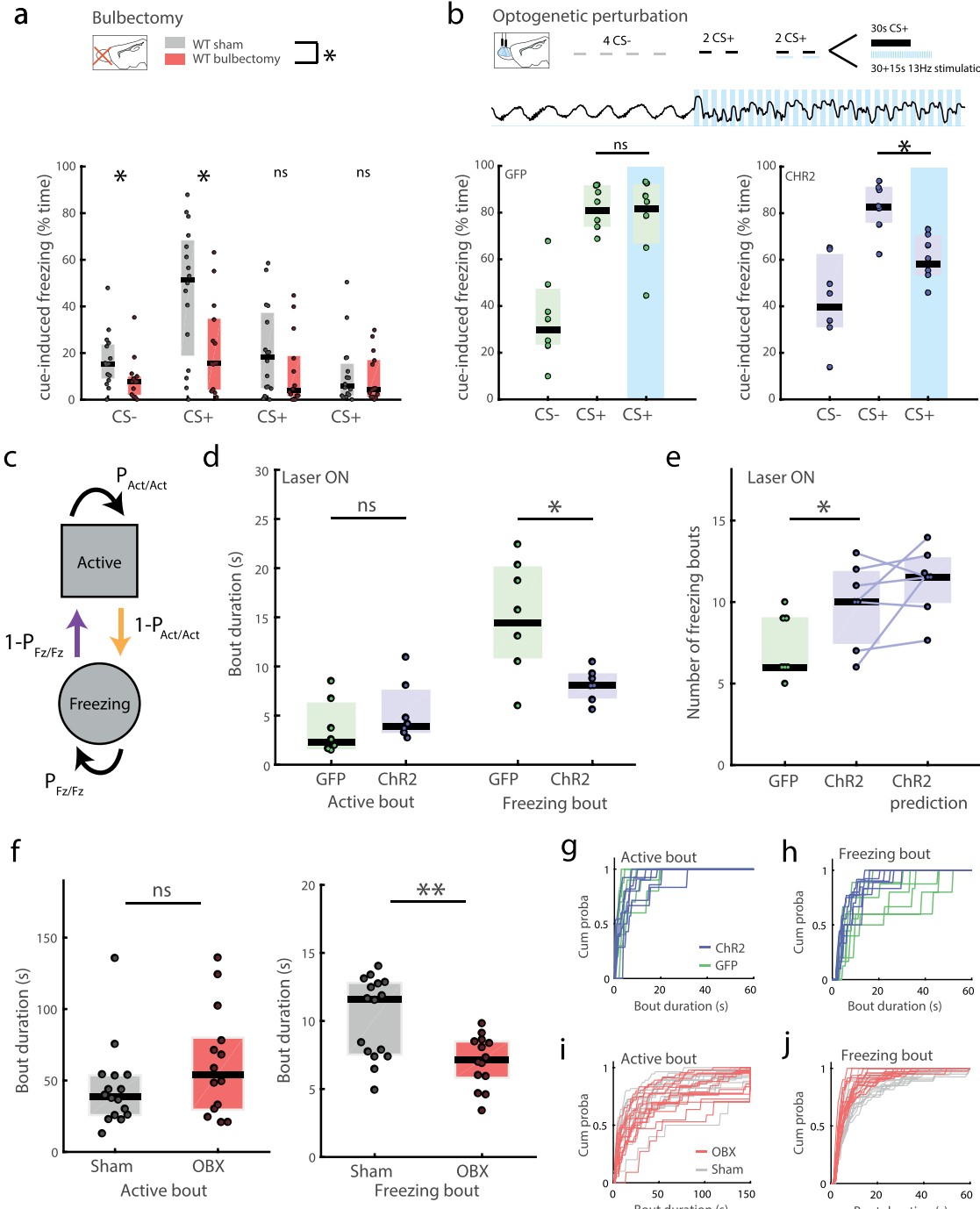

appeared after 3 weeks (Fig. S2B, C). Moreover, partial bulbec-tomies induced none of the detrimental changes observed with total ablation (see Fig. S2 for a full description). The reduction in freezing levels is therefore unlikely to be simply linked to gross behavioural changes associated with bulbectomy. We suggest instead that it can be explained by the lack of transmission of the 4 Hz respiratory-linked oscillatory activity to downstream areas by the OB.

As bulbectomy was performed prior to conditioning, it might have affected both the encoding phase of learning and fear expression. Moreover, despite the controls discussed above, we cannot exclude consequences on neural activity beyond the suppression of the 4 Hz oscillation. To specifically assess the role of the oscillatory output of the OB during fear expression, we

switched to an optogenetic approach and tested whether disrupting 4 Hz during CS+ presentation was sufficient to alter freezing behaviour. We perturbed the temporal organization of the OB 4 Hz activity by rhythmically activating OB interneurons (ChR2 in GAD-Cre mice) at a competing frequency of 13 Hz, to scramble the OB output. We focused stimulation on OB glomerular interneurons since they are responsible for the low-frequency components of OB activity (Fig. S9A)[38]. Perturbation of OB oscillations during the CS+ reduced freezing levels in ChR2 mice (effect size: 2.08) while the same stimulation had no effect on GFP controls (Fig. 3b). Importantly, stimulations prior to conditioning had no effect on the overall behaviour of the mice, excluding a motor confound that could bias freezing detection (Fig. S9B). Stimulation at 4 Hz also produced a

**Fig. 3 4 Hz respiratory rhythm is specifically involved in the regulation of freezing maintenance. a** Median freezing levels of sham and bulbectomized mice during the test session. (Two-way mixed repeated measures anova: group ($p = 0.038$, $F = 4.73$) × CS block ($p < 0.0001$, $F = 23.7$), interaction ($p = 0.0065$, $F = 4.83$). Post-hoc Wilcoxon rank-sum on each block: zval = 2.28, 2.22, 1.53, 0.33; $p = 0.022, 0.026, 0.1239, 0.73$, $n = 16, 14$; effect size for significant differences: 0.77, 0.97). **b** Median freezing levels of GFP- and ChR2- mice during the test session. (Wilcoxon signed-rank test: Signed Rank statistic = 16, 28, $p = 0.813, 0.016$, $n = 7, 7$; effect size for ChR2 with/without stimulation: 2.08; effect size ChR2 vs GFP during stimulation: 1.2). Top: OB LFP raw trace (1, 75 s) at onset of laser stimulation. **c** Markov model of freezing behaviour. At each time step, depending on the current state (Fz or Act), the next state is randomly selected according to fixed probabilities. **d** Duration of active and freezing bouts during 13 Hz laser stimulation for ChR2 and GFP mice. (Wilcoxon rank-sum test: rank-sum value = 42, 71,: $p = 0.2, 0.017$; $n = 7, 7$). **e** Number of freezing bouts during 13 Hz laser stimulation in the test session. The number of freezing bouts for ChR2 mice predicted by Markov chain model is obtained using $P_{Fz/Fz}$ fitted to laser stimulation data but keeping $P_{Act/Act}$ at the laser-off value. This shows that the change in $P_{Fz/Fz}$ is sufficient to predict the increase in freezing bouts. (Wilcoxon rank-sum test GFP vs ChR2: value = 36.5: $p = 0.037$; $n = 7, 7$). **f** Duration of active and freezing bouts and frequency of freezing bout initiation for sham and OBX mice. (Wilcoxon rank-sum test: rank-sum value = 219, 312: $p = 0.23, 0.0083$; $n = 16, 14$). **g**, **h** Cumulative distribution of active (**g**) and freezing (**h**) bout lengths during CS+ presentations during 13 Hz laser stimulation. Note the upwards shift of the freezing bout duration curve for ChR2 relative to GFP mice and the disappearance of episodes longer than 30 s. See Fig. S9C for laser-off distributions. **i**, **j** Cumulative distribution of active (**i**) and freezing (**j**) bout lengths during the whole test session. As in (**g**, **h**) active bout durations are similar but freezing bout duration is reduced. In all panels, boxplots show median and interquartile range.

scrambling effect on the OB oscillation and led to a decrease in freezing (Fig. S11).

The results of these two experimental approaches complement each other because optogenetics restricts manipulation to fear expression whereas bulbectomy removes OB 4 Hz oscillations while avoiding stimulation-induced perturbation of regions downstream of the OB. Taken together, they both clearly point to the same conclusion that OB 4 Hz is necessary for freezing expression.

Our results are at odds with previously published results showing instead that destruction of the olfactory epithelium using the anti-thyroid drug methimazole increases freezing[23]. In order to understand the discrepancies between this experiment and our own, we carefully studied the impact of this drug on fear behaviour. We first injected the drug at the same dose as in ref. [23], successfully reproduced the uncoupling of OB activity from respiration and observed a similar trend towards an increase in freezing after CS+ presentation as in[23] (Fig. S10A, B).

However, a number of control experiments make these results difficult to interpret. First and foremost, we observed that methimazole deeply affected general physiology, inducing substantial weight loss, REM sleep suppression, reduction in body temperature and marked changes in breathing (Fig. S10D–I). This is consistent with methimazole's modification of thyroid activity which will, in turn, modify central metabolism. Given these observations, we hypothesized that methimazole could affect overall activity levels. Therefore the drug might appear to increase freezing without having any specific effect on fear-learning or expression itself. To assess this, we reproduced the protocol used in ref. [23], but without any shock or tone delivery. We found an increase in freezing levels during the latter part of the behavioural sessions, referred to as 'post-tone' period, but not during the early 'pre-tone' period, exactly as in[23] (Fig. S10C). The observed immobility is, therefore, likely to be non-specific to the fear-conditioning paradigm. Based on these findings, it is likely that the increase of freezing after methimazole treatment cannot be clearly interpreted as the consequence of uncoupling the OB from breathing, but may be due to other non-specific behavioural changes.

To conclude, both the results of the bulbectomy and optogenetic manipulation showed that impairment of OB function reduced freezing levels, in contrast to other methodologies that are biased by non-specific effects (see 'Discussion' for full comparison). Taken together, they demonstrate that the 4 Hz respiratory rhythm, transmitted via the OB, is necessary for fear expression.

### 4 Hz rhythm regulates freezing maintenance independently of initiation. Our results manipulating the OB 4 Hz which is

transmitted to the dmPFC agree with results directly manipulating 4 Hz in the dmPFC[25,26]. This raises the possibility that we could use this approach in order to shed light on the exact role of the dmPFC in freezing regulation. We reasoned that it would require taking into account the dynamical regulation of this behaviour. So far in the literature, freezing has generally been viewed as a unitary phenomenon, implicitly implying that a single brain structure acts both as an on and off switch. Freezing is therefore quantified using the percent of time freezing. A number of other parameters could be measured such as the number of freezing bouts, freezing bout duration and inter-bout intervals. In order to clarify what each of these parameters quantifies and to describe freezing dynamics in more detail, we modelled freezing using a two-state Markov chain (Fig. 3c).

According to this model, at each time step, the animal is either in the freezing (Fz) or active (Act) state, and randomly transitions to one of the two states according to the probabilities $P_{Fz/Fz}$ and $P_{Act/Act}$. $P_{Fz/Fz}$ is the probability that freezing behaviour will be maintained whereas $P_{Act/Act}$ is the probability of staying in the active state (Fig. S12A). These probabilities can be analytically derived from the easily-measured average duration of freezing and active bouts respectively (Fig. S12A and Supplementary discussion for mathematical derivation). These two parameters control the initiation of freezing and its maintenance/termination independently. Fitting the model to the data, therefore, enables us to assess the possibility that they could be separately controlled by the brain. It should be noted that the Markov model we used to analyze behaviour makes no distinction between freezing maintenance and termination, since both are controlled by the same parameter $P_{Fz/Fz}$: the probability of maintaining freezing is the probability of not terminating it ($1 - P_{Fz/Fz}$).

In order to validate our model, we used the two probabilities ($P_{Fz/Fz}$ and $P_{Act/Act}$) calculated for each mouse to simulate freezing time courses (Fig. S12A). Based on this simulated data, we measured two other variables: the percentage of time spent freezing and the number of freezing bouts. We then compared these variables obtained from simulations to those in the data. For GFP and ChR2 mice as well as sham and bulbectomized mice, the model prediction and data showed an excellent match (Fig. S12B, C), which validates the Markov model as a good description of freezing behaviour.

Using this framework, we analysed the effect of both bulbectomy and our optogenetic perturbation of OB activity. We found that in both cases freezing bout duration was reduced but active bout duration was not changed (Fig. 3d, f). These differences are also clear in the cumulative distributions of freezing and active bout durations (Fig. 3g–j): active bout distributions overlap, whereas freezing bout distributions are

segregated. Using the Markov model formalism, we can say that ablation and optogenetic perturbation of the OB specifically reduce freezing maintenance probability ($P_{Fz/Fz}$). On the contrary $P_{Act/Act}$ that controls initiation is unaffected.

The number of freezing bouts is a parameter that is sometimes used to describe freezing behaviour. Intuitively, one might imagine that a decrease of freezing could be due to either the reduction of the duration or the number of freezing bouts. Our model predicts that a selective drop in the probability of freezing maintenance ($P_{Fz/Fz}$) which makes freezing bouts shorter will lead to more opportunities to initiate freezing. Therefore, in spite of a constant probability to initiate freezing ($1 - P_{Act/Act}$), the total number of freezing bouts may increase. This is exactly what we observed (Fig. 3e, GFP vs ChR2). This leads us to argue that using the number of freezing bouts as an index of freezing may be hard to interpret and that a simple framework such as this Markov model allows for clearer interpretations.

Moreover, this increase in number of freezing bouts can be predicted for each individual mouse by holding constant $P_{Act/Act}$ calculated from the laser-off phase and simply modifying $P_{Fz/Fz}$ (Fig. 3e, ChR2 data vs ChR2 prediction). This clearly shows that a change in the maintenance of freezing induced by the laser stimulation is entirely sufficient to explain the change in behaviour we observe. This further validates the model and supports the existence of two separate brain mechanisms controlling initiation and maintenance of freezing behaviour independently.

Analyses of both bulbectomy and optogenetic-induced changes of behaviour converge in demonstrating that only the maintenance/termination but not the initiation of the freezing state is modified by OB perturbation.

**Strong 4 Hz power is associated with sustained freezing and freezing offset**. The specific role of the 4 Hz rhythm in freezing maintenance/termination should also be reflected in the dynamics of the 4 Hz oscillation and dmPFC single unit activity in relation to freezing episodes. Since the model does not distinguish between termination and maintenance, neural activity should be more tightly associated with the sustained period or the offset of freezing (or both) than the onset.

During a freezing episode, 4 Hz power gradually rises at freezing onset but drops off sharply at offset (Fig. 4a). This indicates that although 4 Hz power varies with both freezing onset and offset, it is most closely associated with episode termination. Indeed, between individual animals, offset dynamics were more robust and reproducible than onset dynamics (Fig. 4b, c). Accordingly, 4 Hz power was a better predictor of freezing vs active state around offset than around onset (Fig. S13A, B). This asymmetry between freezing onset and offset cannot be explained by a difference in the behaviour of the animal, even when using the highly sensitive measure of head acceleration (Fig. 4d, e).

We found evidence that strong 4 Hz is also associated with more stable freezing. The CS+ was formed of a sequence of short pips that sometimes elicited micromovements detectable using head acceleration. We found that these micromovements occurred most frequently at freezing onset, when 4 Hz was at its weakest and become less frequent as 4 Hz power increased (Fig. 4f). Moreover, their amplitude negatively correlated with the ongoing 4 Hz power (Fig. 4g). This suggests that strong 4 Hz is associated with freezing that is less sensitive to external perturbation and therefore likely more stable. Accordingly, strong 4 Hz was associated with longer freezing episodes. First, long freezing (>10 s) episodes were associated with higher 4 Hz power than short episodes (Fig. 4h). Moreover, at the level of single freezing bouts, we found that 4 Hz power correlated well with the freezing bout length (Fig. S14).

Altogether, we found that 4 Hz power was more tightly associated with freezing offset than onset and that strong 4 Hz power is associated with longer, more stable freezing episodes. This strongly corroborates the hypothesis that 4 Hz is involved in freezing maintenance and termination but not initiation.

**dmPFC single units entrained by 4 Hz are strongly active during sustained freezing and offset**. dmPFC single units showed strong modulation of their activity in relation to freezing: half of the recorded units showed significant modulation of their activity to freezing, 75% of which reduced their firing rate (Fig. 5a). Principal component analysis showed that this modulation was composed of a sustained change of activity throughout the episode (PC1) and of a phasic response at freezing onset and offset (PC2) (Fig. 5b). We, therefore, chose to separate the dmPFC population into three groups: (1) the 'sustained-off/transition-on' units that showed decreased firing throughout the freezing episode but increased their activity at freezing onset and offset, (2) the 'non-responsive' units and (3) the 'sustained-on/transition-off' units that increased firing during freezing and showed transient inhibition at onset and offset (Fig. 5d–f, i). The two groups of responsive units (1 and 3) shared strikingly similar response dynamics but with opposite sign. In particular, they both showed significantly deeper modulation at freezing offset than onset (Fig. 5d–f, ii). Moreover dmPFC activity better predicted freezing vs active behaviour at offset than onset (Fig. S13C, D). Thus, dmPFC single unit firing, like 4 Hz power, was more tightly tied to the sustained period and offset of freezing than to onset, consistent with a role in controlling freezing maintenance/termination.

We found that the three dmPFC neuron groups, defined purely by their freezing-related firing, were differentially entrained by the 4 Hz rhythm. First, both freezing responsive groups had higher proportions of 4Hz-entrained units than the unresponsive category (Fig. 5d–f, iii). Second, the units' freezing response was linked to their preferred 4 Hz phase: 'sustained-off/transition-on' neurons tended to fire on the descending phase of the 4 Hz oscillation whereas 'sustained-on/transition-off' neurons preferred the ascending phase (Fig. 5d–f, iv). This analysis was confirmed by using the reverse approach: we separated units depending on 4 Hz oscillation entrainment and evaluated their response to freezing. Entrained units showed a stronger sustained- and offset-response than the non-entrained units but no difference at freezing onset (Fig. 5g, h).

If the OB 4 Hz oscillation is truly instrumental in organizing the freezing-related activity in the dmPFC, perturbation of this rhythm should impact freezing-related firing. Indeed, we found that dmPFC units recorded in bulbectomized animals showed significantly weaker modulation of their firing rates during sustained and offset periods (Fig. 5g, h). On the contrary, the response at freezing onset is unaltered and we observed no change in overall dmPFC firing rates (Fig. S8A). This strongly suggests that the OB 4 Hz oscillation plays a causal role in establishing dmPFC unit responses to freezing specifically related to freezing maintenance and termination.

By what mechanism could 4 Hz contribute to generating the dmPFC sustained- and offset-responses? Our analysis so far has focused on single-unit responses but oscillatory activity is generally thought to coordinate neural populations. We, therefore, evaluated how the dmPFC network activity evolved. We tracked changes in the population vector before, during and after freezing by measuring the correlation between this vector calculated at different time points (Fig. 5i, see Fig. S15 for details of method). As expected from the strong change in firing during freezing, population vectors during freezing and active periods were anti-correlated.

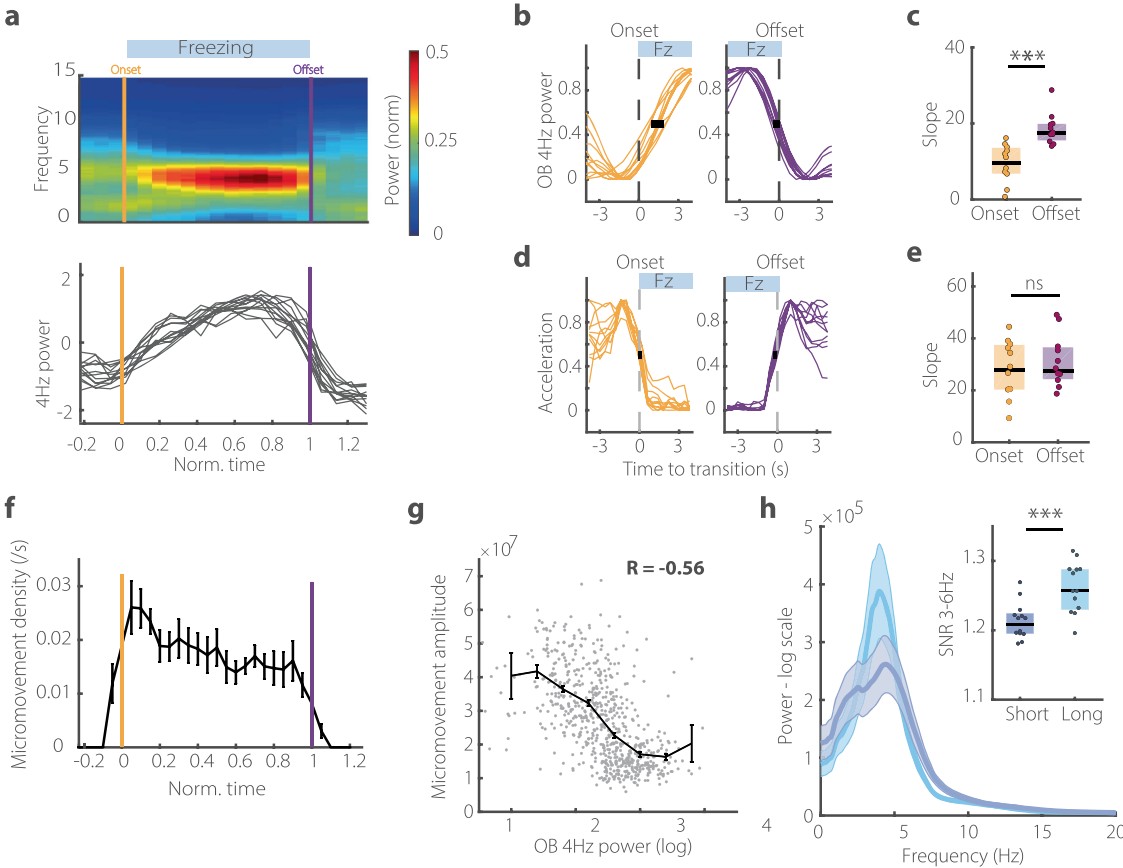

**Fig. 4 Stronger 4 Hz power is linked to more sustained and stable freezing episodes. a** OB spectrogram (top) and mean power in the 4 Hz band (bottom) triggered on freezing periods of normalized duration for all mice. Note the asymmetric rise and fall of 4 Hz power during the freezing bout with a gradual onset and steep offset. ($n = 15$ mice). **b, d** OB 4 Hz power (**b**) and head acceleration (**d**) triggered on onset (left, orange) and offset (right, purple) of freezing bouts. Both measures are normalized between 0 and 1. Black bar indicates the standard deviation at the midpoint. ($n = 11$, 11 mice, only mice with sufficient number of 4 s bouts of freezing are retained). **c, e** Slope of onset and offset curves of and OB 4 Hz power (**c**) head acceleration (**e**) to quantify the difference in transition speed, estimated using a sigmoid fit. (Wilcoxon rank-sum: zval $= -3.61$, $-0.32$; $p = 3e-3$, 0.74, $n = 11$). **f** Density of micromovements during freezing periods of normalized duration showing that they decrease throughout the episode, following a similar dynamic to 4 Hz increase. These small head movements are detectable using head accelerometer only but are not readily visible on video measurements and often coincide with the bips of the CS. ($n = 11$ mice). Error bars are SEM. **g** The amplitude of the micromovements elicited by CS pips is correlated with the instantaneous 4 Hz power. (Pearson correlation: $R = -0.56$, $p = 2.3e-54$, $n = 578$ movements from 11 mice). Error bars are SEM. **h** Averaged OB spectra during short (<10 s) and long (>10 s) freezing periods showing stronger 4 Hz during long episodes. Error bars are SEM. Inset: average signal-to-noise ratio of 3–6 Hz band during short and long episodes. (Paired Wilcoxon signed-rank test: Signed Rank statistic $= 0$, $p = 2.44e-4$, $n = 13$). In all panels, boxplots show median and interquartile range.

During freezing, a clear pattern can be seen in the matrix describing population activity: two blocks of coherent activity emerge during the first third and last-two thirds of freezing respectively. We, therefore, observe two different stable states of population activity in the dmPFC during freezing. The first block corresponds to an initial pattern of population activity at the beginning of the freezing episode. Then, a new pattern appears and remains stable until the end of the episode, when it disappears as the animal becomes active. The second population pattern appears with a similar temporal profile to the increase in 4 Hz power (Fig. 5i, j). This suggests that the 4 Hz modulation of dmPFC units leads to the appearance of a stable dmPFC network activity state during the sustained period of freezing that may participate in its maintenance and/or termination.

## Discussion

In this study we have shown that freezing is associated with regular 4 Hz breathing in mice and that this rhythm, via the OB, feeds back to the dmPFC where it organizes neuronal activity. Both the behavioural effects of surgical and optogenetic

perturbation, as well as analysis of electrophysiological recordings, show that this oscillation supports a specific regulation of freezing maintenance with no effect on initiation. Together this leads us to propose a new model of regulation of freezing and, more broadly, emotive states.

**Respiratory-related OB 4 Hz, transmitted to the dmPFC, promotes freezing**. During freezing, we found that the OB respiratory rhythm is transmitted to the dmPFC, as shown in previous studies[12,23]. Previous experimental studies on the dmPFC itself have shown that impairing this region, in general or specifically via manipulation of local 4 Hz activity, leads to a reduction in freezing[25,28,39]. Combining these two sets of results, one would expect that impairing OB function and, therefore, transmission of 4 Hz to the dmPFC should also lead to a reduction in freezing. However the only study having tested this prediction found the opposite: perturbing the OB increased freezing behaviour[23]. These contradictory findings raised the question of whether the OB 4 Hz really played a functional role in freezing behaviour.

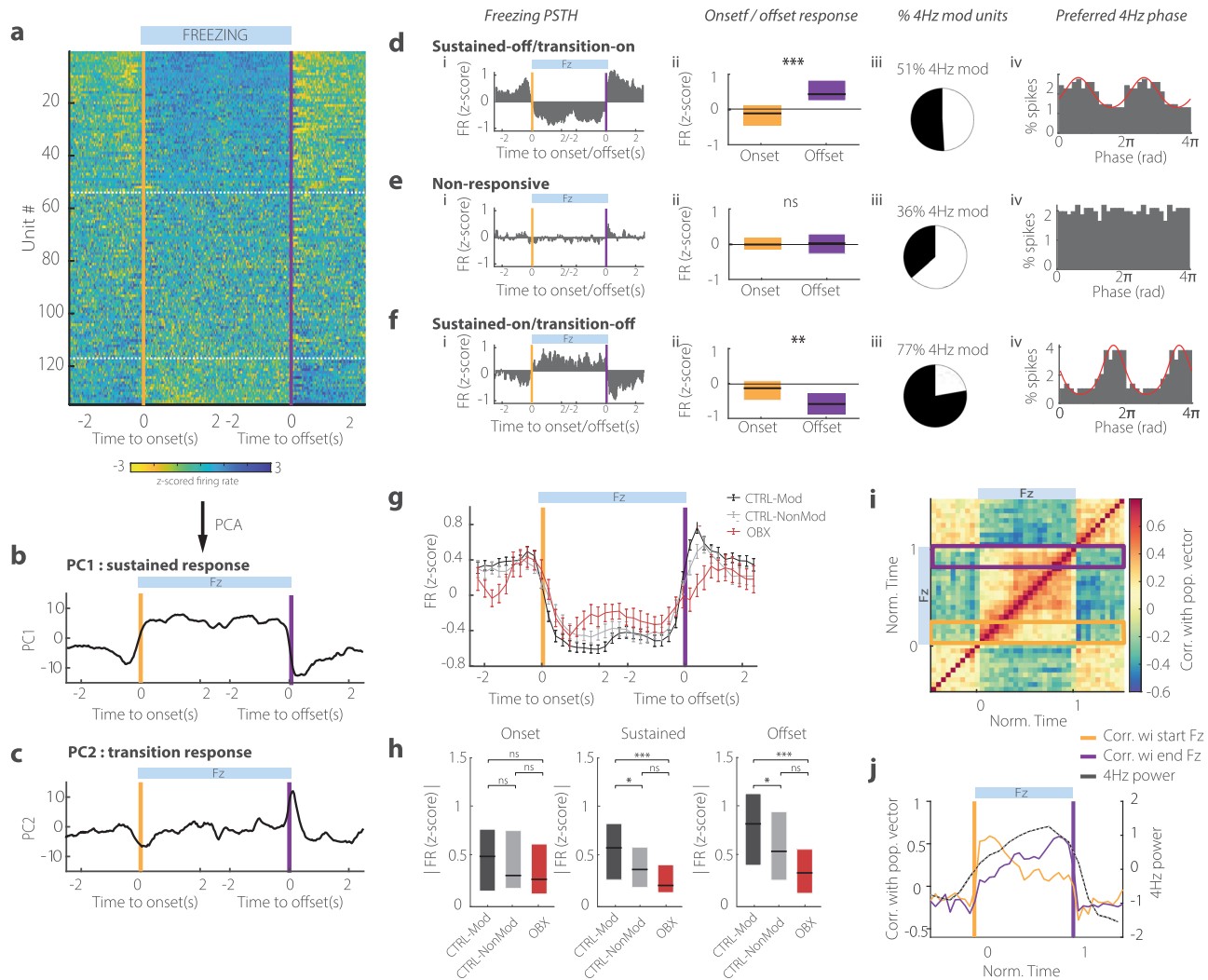

**Fig. 5 Dorsomedial prefrontal units entrained by olfactory bulb 4 Hz display stronger sustained and offset responses to freezing. a** Z-scored response of dmPFC units to freezing onset and offset ordered by average sustained response. White lines delimit the three groups in (**d**, **e**, **f**) ($n = 134$ units). **b**, **c** First two principal components of dmPFC units response to freezing showing the sustained response (PC1: 17% of variance) and the transition response (PC2: 4% of variance). **d–f**: i: Average response of each group of units. ii: Average response at freezing onset and offset. For both groups modulated by freezing, the amplitude of response is stronger at offset than at onset (Wilcoxon signed-rank test: zval statistic = −4.49, −0.65, 2.89, $p = 6.9E−6$, 0.55, 0.0038, $n = 63, 53, 18$). Error bars are SEM. iii: Percent of significantly modulated units by OB 4 Hz. The freezing responsive groups of neurons contain a significantly larger proportion of modulated units. (Chi2 test 57% vs 36%, chi2stat = 6.03, $p = 0.0140$, $n = 71$ & 63 units, 57%: grouped percentage from group 1 and 3). iv: Phase histogram of preferred OB 4 Hz firing phase showing different preferred phase for the two freezing responsive groups (Non-parametric comparison for equal medians: Circ-stat = 5.8, $p = 0.05$). **g** Z-scored response of 4 Hz modulated (black) and non-4Hz modulated (grey) dmPFC units from control animals and of all units from bulbectomized animals (red) during normalized freezing episode. Error bars are SEM. ($n = 91, 43, 29$). All responses were sign-corrected to allow averaging of amplitude. **h** Averaged onset, sustained and offset freezing-responses (absolute value) for modulated, non-modulated and bulbectomized units. (Wilcoxon rank-sum test: Onset: Z-val statistic = 0.12, 1.11, 1.46, $p = 0.9, 0.26, 0.14$; Sustained: Z-val statistic = 2.1, 3.7, 1.68, $p = 0.0002, 0.091, 0.03$; Offset: Z-val statistic = 1.9, 2, 4.8, $p = 7E−6, 0.045, 0.014$, $n = 91, 43, 29$). **i** Temporal correlation matrix of dmPFC population vector before, throughout and after freezing. **j** Average correlation of dmPFC population vector with the population vector at onset (orange box in **i**) and offset (purple box in **i**) and average 4 Hz power (grey) during a normalized freezing episode. Note that the shift from one type of population activity to another closely parallels the change in 4 Hz power.

First, does the OB 4 Hz genuinely entrain local dmPFC activity? Indeed, given the large fields generated by the OB, the LFP recorded in the dmPFC could be due to volume conduction. We found that out of a large population of dmPFC units, 45% were entrained by the 4 Hz rhythm. We demonstrated that the OB is necessary for the transmission of the respiratory rhythm to the dmPFC and that its artificial stimulation at 4 Hz is sufficient to entrain dmPFC firing. Moreover, our results using bipolar derivation and results in[34] using CSD show that the dmPFC LFP is at least partially due to local currents. Furthermore, optogenetic

stimulation of the OB at different frequencies during sleep lead to a nonlinear response profile in the dmPFC LFP, with a very low amplitude at 10 Hz, in contrast to responses at neighbouring frequencies (7 and 13 Hz), a profile that cannot be explained by pure volumic conduction. Altogether these results show that OB 4 Hz is synaptically transmitted to the dmPFC and entrains local activity. The pathway between the OB and the dmPFC is as yet unknown. There is no current evidence of any monosynaptic connections between the OB and the dmPFC. A potential di-synaptic pathway may be via the taenia tecta[23] but given that the

respiratory rhythm has been recorded in multiple cortical areas, the dmPFC may also inherit it indirectly from intra-cortical connections. Overall, although the pathway is unknown, the impact of the OB 4 Hz on local dmPFC activity is clear and strong enough to entrain oscillations of almost half of the neuronal population in the dmPFC.

Second, we found that both bulbectomy and optogenetic stimulation clearly show that reducing the 4 Hz rhythm leads to a reduction in freezing. This is coherent with results directly manipulating activity in the dmPFC which show that dmPFC inactivation reduces freezing levels[25,26,30,39].

This raised the question of why the results of Moberly et al.[23] seem to be in contradiction both with our findings manipulating the OB and findings from the literature manipulating the dmPFC. This study showed that the suppression of OB 4 Hz, via the administration of either TTX to the OB or of methimazole to destroy the olfactory epithelium, increased freezing. Concerning TTX, a non-specific effect on freezing levels can be observed in the original publication (Fig. 6e of ref. [23]) since even before tone presentation the freezing level of TTX mice is three to four times larger than their controls. When converting the raw time count into percentage time, this value is close to post-tone freezing levels of control mice. Concerning methimazole, here we show that it leads to a broad perturbation of overall physiology and that an increase in periods of complete immobility can be observed without any conditioning protocol, indicating that what is quantified as 'freezing' may simply be immobility without relation to defensive behaviour. Given that methimazole is above all an anti-thyroid drug, these global modifications in behaviour and physiology could be attributed to changes in metabolism, in particular to the reduction in body temperature. Indeed, previous reports showed that cold-stressed mice display a drastic reduction of movement[40]. In light of these results, any clear link between methimazole effects and the uncoupling of the OB from breathing is extremely difficult to establish. Therefore, TTX and methimazole protocols both yield non-specific increases in immobility which may not be linked to fear-related freezing but to other motor behaviours. This contrasts with partial bulbectomy and optogenetics experiments presented here, which outside of the conditioning protocol have no impact on motor behaviour.

Taken together, these results show that the suppression of 4 Hz respiratory-related rhythm in the OB leads to a decrease in freezing, just as the suppression of 4 Hz in the dmPFC does. Therefore, the respiratory rhythm originates in the OB, is transmitted to the dmPFC where it organizes neural firing and supports the dmPFC's function in controlling freezing expression. This is consistent with the robust effects of stimulating in and out of phase of the 4 Hz oscillation in the dmPFC in either maintaining or terminating ongoing freezing episodes, as demonstrated in Dejean et al.[26].

Several studies have previously shown that many brain regions, even not directly linked to olfaction, are modulated by the respiratory rhythm[9,11,12,14,17,18] but previous lines of argument mainly suggested that breathing participated in sensorimotor activities such as sniffing, whisking or licking[22]. Here we instead demonstrate that it is involved in organizing activity within the cue fear conditioning circuit and that this participates in sustaining freezing behaviour. Overall, this points to a non-olfactory role for the respiratory rhythm, in agreement with recent observations in humans[18,41].

**Respiratory-related 4 Hz rhythm in the dmPFC controls freezing maintenance**. Taken at face value, the above results suggest that if a mouse begins to breathe at 4 Hz, this will entrain the OB and in turn the dmPFC and, therefore, induce freezing

(i.e. initiate freezing). This would be surprising and seems highly unlikely, in particular since 4 Hz breathing can be observed, albeit less often, outside of freezing. In a similar vein, the role of the dmPFC in top-down inhibitory control seems hard to reconcile with any role in direct control of fear expression. A key question is therefore how we can reconcile, both for the breathing rhythm and the dmPFC, the fact that they can indeed modify freezing expression with their broader functional roles which make it unlikely that they directly pilot freezing behaviour.

The finding that perturbing the OB 4 Hz rhythm specifically reduced maintenance of ongoing freezing episodes, independently of any effect on freezing initiation provides insight into this issue. It leads us to propose a causal chain of emotional generation that involves two major steps. First, behavioural (freezing) and physiological (breathing) responses to the threatening situation are initiated via the classical pathways of the amygdala and periaqueductal grey. Indeed, stimulation of these areas can robustly initiate freezing behaviour[42–44] and amygdala ablation modifies bout initiation[45]. Moreover, these structures are known to modify breathing, possibly via the nucleus retroambiguus[46,47]. Once initiated, the second part of the process is mediated by regular breathing. This rhythm entrains the OB then the dmPFC, allowing to organize neural activity in this region. This likely participates in the neural computations linked to the sustained and offset period of freezing. This is responsible for maintaining long freezing episodes (Fig. 6). Therefore, this second branch of the brain–body–brain loop which relies on somatic feedback (breathing), specifically regulates freezing maintenance. Based on this model, neither the dmPFC nor the OB 4 Hz rhythm directly controls freezing expression but both are necessary in order to sustain long freezing episodes.

**Top-down control of freezing behaviour: relation to global function of PFC**. This two-step model throws new light on the role of the PFC and in particular its dorsomedial subregion in the control of freezing and also on the link between its function in emotional regulation and its broader role in top-down control.

Although the function of the PFC remains much debated, strong evidence supports its role in inhibition both at the cognitive and motor level[48–50]. As a concrete example, in the Wisconsin Card sorting task, patients with frontal lesions who were asked to sort cards according to one rule (colour for example) perform like controls. However, when the rule changes to sort by shape, they fail to inhibit the old strategy and to switch

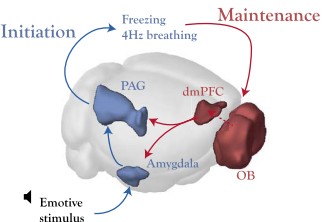

**Fig. 6 Proposed model of brain–body–brain feedback loop of emotional initiation and maintenance.** An emotive stimulus evokes responses in the amygdala and periaqueductal grey (PAG) which lead to changes in behaviour (freezing) and somatic physiology (breathing). The 4 Hz breathing entrains the OB which in turn organizes neural firing in the dmPFC which allows to maintain the ongoing behaviour, perhaps via its connections with the amygdala for example. The dotted line between the OB and dmPFC indicates that this connection is unlikely to be monosynaptic and has not been well described. Image credit: Allen Institute.

to the new strategy. It has been suggested that the multiple forms of top-down regulation provided by the PFC reflect a single common process for behavioural flexibility which may extend to the domain of emotions as well[50,51]. In other words, the maintenance of an ongoing decision-making state could be controlled by a similar mechanism within the dmPFC as the maintenance of a fear state.

In rodents, there has been clear demonstration that the infralimbic cortex is essential for fear extinction, i.e. the reduction of fear-response to a previously threatening stimulus after its repeated non-reinforced presentation[27,52,53]. Since extinction relies on a top-down inhibition of freezing behaviour and is flexible and context-dependent, the role of the infralimbic PFC fits with the global function of the PFC as a top-down regulator of behaviour. The initial studies referred to the syndrome induced by infralimbic PFC lesions as emotional perseveration[27,53]. This highlights the commonality with what has been termed 'para-digmatic perseveration' in reference to the similar lack of behaviour flexibility in cognitive tasks[51].

Contrary to the infralimbic area, the dorsomedial subregion, in particular the prelimbic area, is suggested to be a direct effector for the expression of fear-related freezing itself[26,29]. A direct role for this area in controlling freezing jars with our current understanding of its role as laid out above. Our results instead suggest that the function of the dmPFC during cue-induced freezing is to regulate the duration of freezing episodes independently initiated by the amygdala-PAG system. Therefore, the dmPFC area may not pilot the decision to freeze but instead may determine whether the current coping strategy will be continued or interrupted. This would reconcile the role of the dmPFC in emotional regulation with its role in the flexible change of strategy in rule-switching tasks. These results suggest that the role of both infralimbic and dorsomedial areas could be seen as complementary facets of freezing regulation and support the idea of a close link between PFC function in emotional and cognitive flexibility.

It is important to point out that the Markov model we used showed that 4 Hz perturbation was linked to the decrease of $P_{Fz/Fz}$ and this parameter is agnostic about whether 4 Hz perturbation impacted a mechanism that actively maintains ongoing freezing episodes (maintenance) or a mechanism which controls the time at which freezing ends (termination). We found evidence for both possibilities in dmPFC units. 4Hz-entrained dmPFC units showed a stronger sustained change in firing during freezing episodes and an increased offset response, both reduced in bulbectomized animals. The degree to which the dmPFC participates in the two possible mechanisms, therefore, remains an open question, that will require future analysis and manipulation of the 4 Hz oscillation. It has already been shown that stimulating in and out of phase of the 4 Hz oscillation in the dmPFC with optogenetics during a freezing episode can induce a robust maintenance or early termination respectively[26].

**Regulation of freezing by a brain–body loop: relevance to James–Cannon debate**. The results presented here also have a broader relevance to the link between brain and body in emotion which has been discussed since the original James–Cannon debate[54–56]. The debate stems from the clear evidence of numerous and strong correlations between changes in somatic physiology and emotional state[6,56,57]. However, the key question under dispute is whether somatic changes associated with emo-tion are merely the expression of an emotional state or if they are relayed back to brain circuits to play an active role, in particular, in generating the subjective feel of a particular emotion. The latter position, often referred to as the James–Lange theory, is

supported by evidence of the bodily state influencing various aspects of emotional cognition[58–60] but the exact nature of this bodily feedback and the mechanisms underlying its impact on neural processing remain to be elucidated.

A key weakness of the James–Lange proposal is the dearth of evidence that the suppression of bodily feedback impacts emotions. Indeed, Cannon's original criticism of the James–Lange proposal strongly depended on his finding that removal of the sympathetic system did not produce a reduction in emotional behaviour[61]. Similarly, human studies of spinal cord lesion patients have produced mixed results with some studies not finding any clear changes[62–65].

The present interventional study brings two main contribu-tions to this controversy. First, we demonstrate that blocking bodily feedback modifies emotional states and therefore provide new support for the James–Lange theory. Second, the model of emotional regulation that we propose can provide a possible explanation for the previous observations concerning emotions even in the absence of bodily feedback. Indeed, in our model, although of course the brain initiates a change in behaviour, it is the feedback of the bodily expression of the emotion (breathing) which acts to sustain its expression.

Therefore, there are aspects of emotional behaviour that are independent of bodily feedback (here, initiation), as proposed in the Cannon–Bard theory. These will be preserved after visceral or spinal lesions. Moreover, in such pathological cases, remnant emotional experience might still emerge from repeated initiation events such as during a long-lasting, continuously presented fearful stimulation. On the other hand, when bodily feedback is present, it allows to increase the duration of the emotional state (both at the central and peripheral level). This prolongation could be essential for allowing the state to become a full, conscious emotional experience. For example, if the aversive stimulation is brief, then the fearful state must be maintained by a process dependent on bodily feedback. This leads to the clear prediction that careful investigation of human emotion under privation of bodily feedback will reveal that the remnant emotional experi-ences are of shorter duration. A careful reading of verbal reports from spinal cord lesion patients indeed provides some support for this and suggests this prediction may be borne out by careful testing (see Supplementary Discussion).

To conclude, our results provide both a mechanism by which bodily feedback can impact emotional processing and a demonstration that this feedback can causally impact emotional behaviour. This supports the ideas of embodied cognition[66–68], by exhibiting a mechanism by which somatic physiology becomes an integral part of neural processing. Moreover, we provide a framework for reinterpreting the divergences between the James–Lange and Cannon–Bard theories of the role of the body in emotions. In this framework, the umbrella term 'emotion' must therefore be parsed into three sequential facets: a subcortical emotion leads to a bodily emotion that must feedback to the cortex for this state to last long enough to become a full cognitive, and perhaps consciously experienced, emotion.

The above-proposed interpretation attempts to reconcile behavioural and physiological data from non-human animals and reports of conscious experiences from humans. However, there remains disagreement on how to link these two approaches, in particular in the case of fear for which there is strong controversy about whether to apply this term to non-human animals where the subjective experience of fear cannot be evaluated[69]. In fact, this issue also contributes to make the original terms of the James–Cannon debate somewhat ambig-uous. James's proposal was specifically concerned with the conscious experience of emotion whereas Cannon was studying the outward manifestations of emotion in non-human animals[70].

Although it has been argued that 'fear' should be reserved for the human experience, here we use the term to broadly refer to both. Indeed, neural activity and pharmacological responses that correlate with emotional behaviour in humans and non-human animals also correlate well with reports of subjective emotional experience. This suggests that effects observed in the behaviour of non-human animals can be speculated to be linked to effects in human subjective experience[69,71,72]. We, therefore, tentatively propose that it may be interesting to study the link between bodily feedback and the duration of emotional bouts as subjectively experienced in humans, as suggested above in reference to spinal cord patients.

In conclusion, our findings show that the respiratory rhythm plays a key role in the maintenance of freezing through its entrainment of dmPFC activity. Therefore, freezing should not be viewed as a monolithic behaviour but instead a response to threat that can be dynamically regulated by at least two neural systems, for initiation and maintenance respectively. Using this new functional separation, we suggest that the role of dmPFC in freezing maintenance, and more generally emotional behaviour, is in fact similar to its broader functions as top-down controller of strategy switching. Finally, these results provide an example of a specific mechanistic role for bodily feedback in emotional regulation. This allows to propose a way of reconciling results related to the James–Cannon debate: the role of bodily feedback would be, once an emotion is initiated, to prolong it sufficiently to allow for a full, conscious emotion.

## Methods

**Subjects**. Mice were housed in an animal facility (08:00–20:00 light), one per cage after surgery. All behavioural experiments were performed in accordance with the official European guidelines for the care and use of laboratory animals (86/609/ EEC), in accordance with the Policies of the French Committee of Ethics (Decrees n° 87–848 and n° 2001–464) and after approval by ethical committee (reference: 2016-09). Animal housing facility of the laboratory where experiments were made is fully accredited by the French Direction of Veterinary Services (B-75-05- 24, 18 May 2010). Animal surgeries and experimentations were authorized by the French Direction of Veterinary Services for K.B. (14-43).

**Electrode implantation and virus injection**. C57Bl6 male mice between 3 and 6 months old were implanted under deep anesthesia with a xylazine (10 mg/kg)— ketamine (100 mg/kg) mixture. Electrodes were placed in the right olfactory bulb (AP +4, ML +0.5, DV −1.5), the right CA1 hippocampal layer (AP −2.2, ML +2.0, DV −1.0) and the right prefrontal cortex (AP +2.1, ML +0.5, DV −0.5).

A total of 22 GAD-Cre-line mice were injected with AAV5-ChR2 virus (AAV-EF1a-DIO-hChR2(H134R)-EYFP, 15 mice) and control AAV5-GFP virus (AAV-EF1a-DIO-EYFP, 7 mice) in bilateral OBs. Multiple injection sites were used to obtain optimal expression in the glomerular layer (Table 1) since these interneurons are thought to generate the OB slow oscillations whereas the granular layer interneurons are involved in OB fast oscillations[38] (Fig. S9A).

Virus injections were performed with a glass capillary at a rate of 150 nL/min and followed by a 10 min pause. Three weeks after injection, mice were bilaterally implanted with home-made optic fibres (200 μm-diameter)[73].

During recovery from surgery (minimum 1 week) and during all experiments, mice received food and water ad libitum.

**Bulbectomy**. Animals were randomly assigned to the bulbectomy and sham-operation groups. After drilling, the bulbs were aspirated with a micro-pipette. The cavity was packed with bonewax, the holes were covered with dental cement and the skin sutured. Sham-operated mice were treated similarly, except that the holes were not drilled. After the surgery, animals were housed one per cage. The extent of bulbectomy was measured by photographing whole brains after perfusion and manually measuring the area of the OB on photographs as shown in Fig. S2.

**Electrophysiological recordings and spike sorting**. Signals from all electrodes were recorded using an Intan Technologies amplifier chip (RHD2216, sampling rate 20 KHz). Local field potentials were stored at 1250 Hz. Analyses were performed with custom made Matlab programs, based on generic code that can be downloaded at www.battaglia.nl/computing/ and http://fmatoolbox.sourceforge. net/.

Local field potentials were recorded using tungsten wires with PFA isolation (0.002" bare) and single units using home-made tetrodes formed by twisting

### Table 1 Injection sites and virus quantities.

| | | |
|---|---|---|
| 7 ChR2 | Ventromedial AP +5.2, ML 0.5, DV 2.1 | 300 nL each |
| 7 GFP | Dorsomedial AP +5.2, ML 0.5, DV 1.2 | |
| | Ventrolateral AP +5.2, ML 1.7, DV −1.7 | |
| | Dorsolateral AP +5.2, ML 1.7, DV −0.5 | |
| 4 ChR2 | Ventromedial AP +5.2, ML 0.5, DV 2.1 | 200 nL each |
| | Dorsomedial AP +5.2, ML 0.5, DV 1.2 | |
| | Ventrolateral AP +5.2, ML 1.7, DV −1.7. | |
| 4 ChR2 | AP +4.5, ML +0.8, DV −1.6 | 600 nL each |

insulated Nichrome Wire (0.001" bare). Tetrodes were plated to reduce impedances to around 100 kΩ using gold solution (Neuralynx).

For spike sorting, extracted waveforms were sorted via a semi-automated cluster cutting procedure using KlustaKwik and Klusters (http://neurosuite.sourceforge. net/) Recordings were visualized and processed using NeuroScope and NDManager (http://neurosuite.sourceforge.net/).

**Fear conditioning**. On day 1, mice were habituated to the testing environment without sounds. On day 2, mice were submitted to a habituation session in the conditioning context during which the CS− and the CS+ were presented 4 times. Discriminative fear conditioning was performed on the same day by pairing the CS + with a US (1-s foot-shock, 0.6 mA, 8 CS+ US pairings; inter-trial intervals, 20–180 s). The onset of the US coincided with the offset of the CS+. The CS- was presented after each CS+ US association but was never reinforced (8 CS− presentations; inter-trial intervals, 20–180 s). On day 3, conditioned mice were submitted to a test session during which they received 4 and 12 presentations of the CS − and CS+, respectively. The conditioned sounds consisted of a 30 s sequence of 50-ms pips repeated 27 times at 80 dB sound pressure level. The pips of the CS− and CS+ could be either 7.5 kHz or white-noise and were counterbalanced across experimental groups. Different environments were used for conditioning and testing to avoid any contributions of contextual conditioning, they differed in shape (cylinder vs square), colour (black vs white), floor (metal grid vs plastic) and odour cues (70% ethanol vs 1% acetic acid). Fear conditioning experiments were performed using Imetronic fear conditioning system.

To score freezing behaviour, animals were tracked using a home-made automatic tracking system that calculated the instantaneous position of the animal and the quantity of movement defined as the pixel-wise difference between two consecutive frames. The animals were considered to be freezing if the quantity of movement was below a manually-set threshold for at least 2 s. Freezing behaviour was quantified during blocks of CS by calculating the percentage of time spent freezing during the sound and the subsequent 30 s. Animals with electrophysiological recordings were equipped with a headstage (Intan RHD2132 board) which includes as 3-axis accelerometer which gave access to the animal's acceleration with high temporal resolution.

Plethysmography Respiration was recorded with a whole-body plethysmograph (Emka Technologies) with continuous air flow. The pressure signal was amplified via an omnetics headstage and recorded using the same Intan system as the electrophysiological signals to insure synchrony. Breathing frequency is defined as the inverse of the time between two inspirations and the tidal volume is the integral of the negative signal. The system was calibrated to convert voltage fluctuations into mL/s by manually injecting known volumes of air at different speeds.

**Optogenetic stimulation**. Fear conditioning experiment: On day 1, during the habituation session, mice received 13 Hz blue light stimulations (45 s). The stimulation intensity was chosen to see clear LFP oscillations in the OB and the PiCx, an output structure of the OB. No stimulation was delivered on day 2, during the habituation to sounds and the conditioning phase. During the test session on day 3, mice first received 3 short bouts of light stimulation (10 s, inter-stim interval 10 s) before the onset of the first CS. No stimulation was delivered during the CS− and the first pair of CS+. 13 Hz blue light stimulations were delivered during subsequent CS+ (45 s, locked at the CS+ onset, i.e. extending 15 s after the end of each sound).

Sleep stimulation experiment: after an initial light-phase day of baseline recording in their homecage to habituate to sleeping with electrophysiological and optic cables, mice were tracked using homemade Matlab software to identify periods of prolonged immobility corresponding to sleep. During these periods, periodic stimulation of ChR2 was delivered at a randomly selected frequency taken from 1, 2, 4, 7, 10, 13, 15 or 20 Hz. Stimulation was performed with 50% duty cycle. Stimulations were separated by at least 5 min. Light intensity was calibrated to be at 15 mW/mm². 

All experiments used 473 nm light, delivered using a Doric Lense rotary joint and patch cords to a fibre of 0.37 numeric aperture.

**Methimazole injection and behavioural study**. Mice were injected intraperitoneally with 75 mg/kg of methimazole (Sigma) in sterile saline or an equivalent

volume of saline. After injection, mice were weighed daily and recorded in their homecage all day to assess sleep behaviour. We performed a pseudo-conditioning protocol that imitated exactly the experimental timeline of the methimazole experiment in ref. [23] without shocks or sounds to assess non-specific effects of methimazole administration. Mice were placed for 15 min in a small open field (21 × 30 cm) prior to injection and then were placed in a different open field for 15 min 4 days after injection.

**Histological analysis.** After completion of the experiments, mice were deeply anesthetized with ketamine/xylazine solution (10%/1%). With the electrodes left in situ, the animals were perfused transcardially with 50 mL saline, followed by 50 mL of PFA (4 g/100 mL). Brains were extracted and placed in PFA for postfixation for 24 h, transferred to PBS for at least 48 h, and then cut into 50-µm-thick sections using a freezing microtome. Slides were mounted and stained with hard set vectashield mounting medium with DAPI (Vectorlabs). Location of electrodes was identified and mapped onto the stereotaxic atlas (Paxinos and Watson, 2004).

**Spectral and coherence analysis.** Electrophysiological data were analysed with custom-written Matlab (The MathWorks Inc., Natick, MA, 2000) codes that made use of the Chronux toolbox (http://www.chronux.org). Spectrograms and coherograms were calculated with multi-taper Fourier analysis to diminish finite windowing effects. For all analysis, we applied 5 tapers and a window size of 3 s with a 0.1 s shift between bins. Changes in power of specific bands were calculated only after identification of a peak in the power spectrum (3–6 Hz for the respiratory rhythm) in at least one condition using the signal to noise ratio defined as the ratio between power in the band of interest to the power in the rest of the spectrum. To compute average spectrograms, the spectrograms of each mouse is normalized to total power so that each mouse contributes equally to the average.

To quantify the relationship between behaviour (freezing or not freezing) and power in the 3–6 Hz band, we used receiver operator characteristic curves[74]. This analysis quantifies the ability of an ideal observer to predict whether an animal is freezing or not based purely on the instantaneous power in the 3–6 Hz band. In our case, the observer discriminates the two behavioural states by placing a threshold ($z$) on 2s-binned spectral power ($r$) in the 3–6 Hz band above which the corresponding bin is classified as 'freezing' and below which it is classified as 'non-freezing'. The performance of this procedure can be fully determined by two parameters:

$\alpha (z) = P (r > z|FR)$, size or false alarm rate
$\beta (z) = P (r > z|NOFR)$, power or hit rate

Plotting $\alpha$ and $\beta$ for increasing values of $z$ yields the ROC curve and the area under this curve (ROC value) represents the probability that an ideal observer can discriminate between a 2 s period of freezing or no freezing based on the spectral power recoded at that time: it is equal to 0.5 if the spectral power carries no information about the behaviour and equals 1 it is perfectly predictive.

To be sure to have sufficient data for a fair comparison, we restricted our analysis to animals presenting at least 10 s of freezing and 10 s of non-freezing behaviour ($n = 12/15$). When analyzing specifically onset and offset prediction, mice were further restricted for having a sufficient number of freezing bouts longer than 6 s ($n = 11/15$).

**Phase analysis of LFP signals.** Instantaneous phase of the OB LFP during freezing was derived by applying the Hilbert transform to the signal filtered between 4 and 6 Hz. To estimate the instantaneous phase of signals throughout the behavioural session with highly variable frequencies, since the OB respiratory rhythm that varies from 2 to 15 Hz, we applied a method derived from ref. [75] as filtering in frequency bands to estimate instantaneous phase would fail to capture these dynamic changes. The LFP signal was filtered between 1 and 20 Hz and local peaks and troughs were detected. The phase of the signal was then defined by interpolating between the peak (0°) and the trough (180°). To evaluate the quality of the method for capturing the instantaneous variations in the signal's frequency, a test signal was reconstructed using the instantaneous phase and the amplitude of the detected extrema. The mean square error between reconstructed and real signal was calculated to estimate the quality of the reconstruction (Fig. S7C).

**Phase modulation analysis of units.** Only test sessions with more than 10 s of freezing were used in this analysis.

Using the methods described in the previous section, each spike of each unit was attributed its corresponding phase and the distribution of these phases was tested for uniformity. Given the asymmetric shape of OB oscillations, the phase distributions were corrected as described in ref. [76]. To test the significance of phase modulation we used the standard Rayleigh test for uniformity with a threshold of $p = 0.05$. The relevant statistic used for the test is the variance-stabilized $\log(Z) = R^2/n$ where $R$ is the resultant length and $n$ the number of spikes.

When comparing the modulation of units by OB LFP in active and freezing states (Fig. 2k), we corrected for period duration by only analyzing the period of highest activity of corresponding length to the freezing period. We also corrected for changes in firing rate by subsampling spikes using the bootstrapping method in the period with the highest firing rate. We compared the strength of modulation using two parameters: the frequently used concentration coefficient kappa and the

pairwise phase consistency[77] which is an unbiased measure (Fig. 2l, S7D). To estimate the phase preference of dmPFC units to the respiratory rhythm using OB LFPs, we corrected the phase of modulated units by the average phase difference between dmPFC and OB recordings. This allowed us to avoid confounds due to phase changes with depth in the OB.

**Cell type classification.** For each unit, we used three parameters taken from the literature[78,79] to cluster waveforms into two clusters: duration at half-amplitude, trough to peak latency and the asymmetry index (positive ratio of the difference between right and left baseline-to-peak amplitudes and their sum) (Fig. S5C, D). To cluster the units used in the study we grouped them with a larger set of units (629) recorded from other studies in the laboratory to insure more robust clustering. After clustering, we found 74% putative pyramidal neurons and 26% putative interneurons. Putative pyramidal neurons were broader than putative interneurons and as expected had lower average firing rate (5.5 Hz ± 0.9 s.e.m vs 14.7 Hz ± 1.9 s.e.m).

**dmPFC unit response to freezing.** We tested whether dmPFC units were significantly modulated by freezing using the change detection method[80]. The freezing-triggered PSTH for each neuron is z-scored and the cumulative sum is taken. If a unit is responsive then a consecutive series of positive or negative values is expected and will lead do the cumulative sum deviating from zero. We, therefore, ask whether the peak deviation from 0 of the cumulative sum is significantly greater than that expected by chance. This method has the advantage of adapting well to neurons that have either a very strong response in a few consecutive bins or a small response in many consecutive bins, contrary to bin-by-bin testing. To assess significance, a null distribution was constructed by shuffling the order of bins 1000 times before repeating the analysis. We tested the real value against this distribution using a 5% significance criteria.

**Markov model of behaviour.** The two-state Markov model supposes that the animal's behavioural state toggles between an active state and a freezing state, according to two probabilities $P_{Fz/Fz}$ and $P_{Act/Act}$ (Fig. 3c). These probabilities define, depending only on the current state whether the animals remain in that state or switches. Bout duration, expressed in number of time bins, is related to the probability to stay in that state by the relation:

$$\text{Bout dur(Fz)} = \frac{P_{Fz/Fz}}{1 - P_{Fz}} \qquad (1)$$

$$\text{Bout dur(Act)} = \frac{P_{Act/Act}}{1 - P_{Act}} \qquad (2)$$

These relations are derived in the Supplementary discussion. Both parameters of the model can therefore be directly derived from the measured mean bout duration of freezing and active states.

To validate the model, we then simulated the model 100 times, using a time step of 2 s, to estimate two variables that could be derived from the data: total freezing duration and frequency of freezing bout initiations. Unlike average bout duration, these variables depend on both parameters of the model and therefore do not allow to disambiguate which of the two processes is affected by a given manipulation.

**Modulation index.** To evaluate changes in a given parameter X (firing rate, concentration coefficient) at the level of the individual unit, we define the modulation index to compare situation 1 and 2 for each neuron as:

$$\text{MI} = \frac{X_1 - X_2}{X_1 + X_2} \qquad (3)$$

**Statistics.** For each statistical analysis provided in the manuscript, the Kolmogorov–Smirnov normality test was first performed on the data. If the data failed to meet the normality criterion, statistics relied on non-parametric tests. We, therefore, represent the median and quartiles of data in boxplots in all figures, in accordance with the use of non-parametric tests.

For comparison of proportions, the standard chi2 test was used.

Two-way mixed repeated anova was performed using GraphPad Prism, GraphPad Software, La Jolla, California USA.

Ranksum and signed rank: we report the signed rank statistic if the number of replicates is too weak to provide the normal Z statistic.

All statistical tests were two-sided. On all figures, the following conventions are used *$p < 0.05$, **$p < 0.01$, ***$p < 0.001$.

**Reporting summary.** Further information on research design is available in the Nature Research Reporting Summary linked to this article.

## Data availability
The datasets generated during the current study are available from the corresponding author on reasonable request. Source data are provided with this paper.

## Code availability

Analyses were performed with custom made Matlab programs, based on generic code that can be downloaded at www.battaglia.nl/computing/ and http://fmatoolbox.sourceforge.net/.

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

## Acknowledgements

We thank L. Roux and A. Peyrache for discussion and critical readings of an earlier version of the manuscript. This work was supported by the Fondation pour la Recherche sur le Cerveau (AP FRC 2016), by the French National Agency for Research ANR-12-BSV4-0013-02 (AstroSleep) and ANR-16-CE37-0001 (Cocode), by the CNRS: ATIP-Avenir (2014) and by the city of Paris (Grant Emergence 2014). This work also received support under the program Investissements d'Avenir launched by the French Government and implemented by the ANR, with the references: ANR-10-LABX-54 MEMO LIFE and ANR-11-IDEX-0001-02 PSL* Research University and was supported by the ERC consolidator grant (Grant/Award Number: ERC-CoG- 726169-MNEMOSYNE) G. d.L. and M.M.L. were funded by the Ministère de l'Enseignement Supérieur et de la Recherche, France. S.B. was funded by the ENS-Ulm, PSL Research University, the Ministère de l'Enseignement Supérieur et de la Recherche, France and the Fondation Pour la Recherche Médicale, France.

## Author contributions

S.B. and J.M.L. performed the experiments and analysed the data. M.M.L. and G.d.L. did several experiments included in the dataset. C.B. and M.C. helped for the behavioural experiments. H.G. performed the histology. J.M.L developed the optogenetic and chemogenetic (not in manuscript) methodology. S.B. and K.B developed the framework to link results with emotion theory. S.B. and K.B. wrote the manuscript with the help of J.M.L. and C.H.

## Competing interests

The authors declare no competing interests.
