## [Peer Review File · Nature Communications]

Reviewers' Comments:

Reviewer #1:

Remarks to the Author:

The authors are to be congratulated on conceiving a novel project and doing a very interesting study. The studies are creative and well performed. And I believe that the results are important and potentially worthy of publication. But I have some concerns about the scholarship of the manuscript and the way concepts are discussed.

1. It is unusual to see a scientific paper that says "The main focus of this paper is to support this claim" rather than to objectively test the hypothesis in a more neutral way.
2. Later in the introduction it is stated that "the exact role of the PFC in fear-related behavior is still open to debate" but do not offer any evidence to support this claim. I'm not questioning the point, but only asking for citations that would make the statement credible
3. Throughout the word "fear" is used in an uncritical way, ignoring a major debate in the field about the use of mental state terms like fear in relation to behaviors such as those studied. The debate, which was published in Nature Reviews Neuroscience, has received considerable attention and should be acknowledged. This is not a suggestion to about how the authors should use the term fear, but instead a suggestion that the controversy/discussion be recognized rather than ingored.
4. Towards the end, it is stated: "This model clarifies the role of the PFC in fear related behaviour" and especially its role in behavioral "perseverance." Please search PubMed to see studies that have explored the role of infralimbic and prelimbic cortex in "emotional perseveration" in the context of Pavloivan extinction.
5. Finally: "Our proposed model supports the James-Lange theory" because "emotional breathing entrain oscillations in the brain which in turn play an active role in regulating emotional expression." The James-Lange Theory is not about the regulation of emotional expression but about the nature of conscious emotional feelings like fear. The last sentence comes around to conscious emotion, and implies that the results reported might also explain this. But to bring conscious emotional experience up requires a critical discussion of the emotional experience literature, and also brings us back to the issue about what fear is mentioned above.

These conceptual points are not trivial. The paper has the potential to have an important impact, and it's conceptual content should be as rigorous as the empirical work.

I would be in favor of publication if these points are addressed.

Reviewer #2:

Remarks to the Author:

In this study Bagur et al. describe a role for olfactory bulb (OB) 4 Hz oscillation in maintenance of cue-induced freezing behavior in mice. They also describe a functional connection between the OB and the prefrontal cortex (PFC), which has a preferential frequency of 4 Hz. While providing very interesting results, the study lacks important controls and the results are over interpreted.

First the authors do not provide evidence for a causal role of the 4Hz OB oscillation in cue-induced-freezing. To show causality of OB 4 Hz oscillation, the authors should apply a 4 Hz optogenetic stimulation during the CS-. If this stimulation increases freezing, it would demonstrate the sufficiency of OB 4 Hz oscillation to increase freezing. Indeed, by inducing a 13Hz stimulation they decrease freezing, which could be through alteration of the 4Hz oscillation, but also through alteration of AP

distribution and information flow to the downstream region independently of the 4 Hz oscillation. Similarly, the role of the 4Hz OB oscillation in maintenance rather than initiation of cue-induced freezing, need further demonstration. For example if the authors induce a 4 Hz oscillation in the OB using optogenetics after initiation of a cue-induced freezing bout, the authors should see an increase of freezing bouts length.

Second, the authors argue that the OB and PFC are connected, but do not even hypothesize about the pathway potentially linking these two brain regions. If the author claim a direct connection, as presented in Figure 5, they need to provide anatomical evidence for this monosynaptic connection using optogenetically assisted circuit mapping, anterograde/retrograde tracing, or electron microscopy. If present, the role of this potential direct connection should be tested using direct manipulation of the OB-PFC neurons. Moreover the extreme vicinity between the two regions is a major confound regarding their conclusions and as they admit in the discussion, volume conduction is very likely partially responsible for the 4Hz OB PFC coherence. The proportion of the volume conduction contribution and the proportion of the of circuit connectivity contribution to the PFC 4Hz oscillation need to be discussed more carefully.

Third, in the result section the authors interestingly introduce and discuss previous studies that performed pharmacological inhibition or OBX. However, this section of the manuscript is not structure and these seminal studies need to be cited in the introduction of the manuscript, the results they obtain to reproduced the findings, should be described in the results section (as they are currently) and the discussion of their data in light of the previous studies should be in moved to the discussion section.

Overall, the study needs to be completed to support the claim of the authors, and/or the interpretation of the data need to strongly be tempered.

--- Minor Comments ---

The manuscript contains many typos and would benefit to be read by a native English speaker.

--- Abstract

Second sentence: add a coma after 'Here we report that in mice'.

The last sentence of the abstract is not understandable without reading the manuscript.

--- Results

Entire section: When the authors refer to cue-induced freezing, it would be helpful to specify it.

First paragraph: please rephrase the description of the CS-, currently described as an unconditional stimulus. While this is technically true, this is confusing as the US is usually referring to the electric shock.

Third paragraph: Replace 'at time of testing which.' with 'at time of testing.'

Fourth paragraph 13 HZ = 13 Hz

Sixth paragraph: replace 'we exactly reproduced of the protocol' with 'we reproduced the protocol'

---Figures

- The first figure cited in the manuscript is the Figure 5. Please either move this Figure as Figure 1 or change the text.

- Figure 1B: please add an amplitude scale bar for the oscillation in OB and PFC. An inset with only 3 cycles enlarged would be helpful to visualize the oscillatory dynamics.

- Figure 1C: what is counted ? cycles ? also, please add a legend of the colors on the figure and in the figure legend

- Figure 1 I-J: add y axis units = Cue-induced freezing (% Time)

- Figure S1I: add y axis units = Context-induced freezing (% Time), and CHR2 = Chr2

--- Materials and methods

- Electrode implantation and virus injection – second paragraph
- Replace 'to obtain optimal to the glomerular layer' with 'to obtain optimal expression in the glomerular layer'
- Same sentence: remove 'to' in 'whereas to the granular'
- Fear conditioning: last sentence replace 'they different' with 'they differed'
- Optogenetic stimulation
- 3 shorts bouts = 3 short bouts
- What is the diameter of optic fiber used ? 300 um ?

--- Discussion

- In the second paragraph of the discussion, the authors write '10Hz oscillation we evoke in the OB activity that does not lead to PFC LFP modulation whereas neighboring frequencies of 7Hz and 10Hz do elicit a response' : the authors probably mean 'frequencies of 7Hz and 13Hz do elicit a response' ? Also, the authors should specify the Figure panel they refer to at the end of this sentence.
- Please correct the use of the verb 'to argue': Individuals argue, data do not argue, but suggests
- Do not use double superlative such as 'more strongly' (stronger is the adapted term)

Reviewer #3:

Remarks to the Author:

Bagur et al. investigates the mechanistic underpinnings of the 4Hz oscillation present during freezing in mice. They find that respiration has an effect on freezing maintenance through an OB->PFC circuit. The authors conclude that this phenomenon represents a brain-body-brain loop.

Bagur et al. found that both bulbectomy and optogenetic perturbation of OB activity reduced freezing. This is at odds with findings of Moberly et al., who found that TTX and methimazole in the OB caused prolonged freezing. Bagur et al. compared the different approaches and concluded that TTX and methimazole may have more non-specific, off-target effects. They also present correlative evidence that the respiratory rhythm is linked to termination of freezing more than initiation, and present a simple Markov model consistent with the idea that the key factor influenced by breathing is freezing maintenance.

Next, the authors went on to show that OB influences freezing probably through PFC. This was based on coherence of oscillations, resonance properties probed by optogenetic stimulation and phase locking analysis of PFC units.

The authors made a thorough comparison between different modes of perturbing the OB, all having different limitations. I would be inclined to believe that the techniques presented here provide a more reliable and potentially more interesting, detailed picture on how the OB oscillation influences freezing bouts specifically. At the same time, I feel this bit of information, although interesting to experts of freezing behavior in mice, is somewhat incremental and, especially, a bit detached from the sweeping conclusions the authors tend to make in the manuscript.

The second half of the results, concerning PFC are also interesting, however, the OB->PFC connection is not a novel finding. In this regard, my opinion is that the authors do not properly credit Moberly et al., who reached similar conclusions about this particular connection, illuminating the origins of the 4Hz oscillation that had previously been considered enigmatic.

Taken all of these together, Bagur et al. present interesting, if rather specific and somewhat descriptive new results. However, these seem to be over-interpreted without proper credit assignment. In addition, the manuscript is hard to follow, not very logically built, and could use a thorough re-writing. For instance, the main conclusion of a brain-body-brain loop is an old concept

and has long infiltrated clinical practice at a wide range of medical fields. Therefore, in the present form, I am afraid that the manuscript would not really engage a broad readership. At the same time, these results, after careful re-writing, could be important to the fear learning field.

At this point, as the presentation quality makes it hard to make a detailed assessment, I can only give examples of specific points, below.

- Introduction, para 3. Proposing that investigating freezing initiation and termination separately 'addresses the role of PFC in fear-related behavior' is a gross overstatement.

- There are many broken sentences, typos (e.g. 'mirroring our firing', 'regulate of freezing maintenance', 'tuned respond optimally', 'make it a promising candidate'), lab jargon and oversimplified statements. Without clear writing, the paper becomes extremely hard to process.

General remarks

We would like to begin by thanking the reviewers for their stimulating comments. We feel that they have led us to greatly improve and clarify our manuscript. Thanks to these changes, we provide stronger evidence for and a more focused and informed discussion of our main message that we briefly recap here.

The key message is that during freezing the 4Hz respiratory rhythm, transmitted via the olfactory bulb (OB) to the prefrontal cortex (PFC), organizes neural firing in this region and supports this area's specific regulation of freezing maintenance. This result brings new light to three important and broad issues: it demonstrates a functional role of the respiratory rhythm, it allows to propose a way of reconciling the role of the PFC in fear conditioning with this region's broader top-down function and it demonstrates that freezing, one of the most commonly studied behaviors in neuroscience, is regulated by separate initiation and maintenance processes. Regarding the first point, we must emphasize this result is not merely incremental regarding previous literature but in fact resolves the contradictions in previous results regarding the function of the 4Hz oscillation. There is a general agreement that the 4Hz in the PFC originates in the breathing rhythm and is transmitted via the OB. However, previous results manipulating 4Hz in the PFC (Karalis et al, 2016) found increase in freezing whereas manipulating 4Hz in the OB found reduction freezing (Moberly et al, 2018). Previously, there was no explanation for these contradictory results, which prevented reaching any clear conclusions regarding the function of this rhythm. Our results, based on both novel manipulations and on reproduction of previous experiments with extra controls, yield a coherent picture in which manipulation of 4Hz in the OB and in the PFC both show that this rhythm promotes freezing. Moreover we explain the origin of the misleading results of OB manipulation in Moberly et al, 2018.

The reviewers comments have led to three broad changes:

First, both reviewer 1 and reviewer 3's remarks led us to undertake a **thorough rewriting of the manuscript**. On the conceptual side, reviewer's 1 remarks have allowed us to more carefully situate our results in the broad context of the study of emotions in animals and humans. Reviewer's 3 pointed out that the manuscript was sometimes hard to follow. Consequently, we have rewritten the manuscript in order to focus on the main message of the paper. We have thus removed figure 4 describing the frequency-specific tuning of the OB-PFC pathway and the discussion related to frequency-specific communication which was perturbing the main message of our manuscript. We have re-organised the figures to make the reasoning logic easier to follow.

Second, one of the main requests from reviewer 2 was the addition of **new experimental data with stimulation at 4Hz in the olfactory bulb**. First of all, as you will see, we discussed the theoretical implication of this experiment and described why it is not an obvious expectation of our study. Nevertheless, we performed the experiments: the 4Hz stimulation did not result in an increase of freezing as the reviewer suggested. We describe in detail why stimulating at 4Hz without controlling the phase of stimulation perturbs activity and therefore is in fact expected in our framework to decrease freezing as we indeed observe. As we explain below, the only experimental method to test the hypothesis raised by the reviewer would have been to develop a closed-loop stimulation system to perform the optogenetic stimulation in phase with the ongoing endogenous breathing-related oscillation. The extensive development required and the

limitations due to the current health crisis context render these experiments impossible to perform.

Third, in order to further support the main message of our paper, we decided to add a **new part of in-depth analysis of the PFC neuronal activity** during the freezing episodes. To bolster our argument, we made an extensive analysis of neural activity and find that there is a close tie between PFC firing, freezing maintenance and 4Hz activity. This has led to a new figure included in the manuscript (Fig5).

Both the new data and the new organisation of the manuscript allow us to focus on our key message. We believe this work will be of interest not only to the fear community but also the respiratory related rhythm-, brain rhythm- and affective neuroscience communities.

Reviewer #1 (Remarks to the Author):

The authors are to be congratulated on conceiving a novel project and doing a very interesting study. The studies are creative and well performed. And I believe that the results are important and potentially worthy of publication. But I have some concerns about the scholarship of the manuscript and the way concepts are discussed.

1. It is unusual to see a scientific paper that says “The main focus of this paper is to support this claim” rather than to objectively test the hypothesis in a more neutral way.

We agree that this sentence was badly formulated. The introduction has now been rewritten with a more objective stating of our aims and hypothesis.

2. Later in the introduction it is stated that “the exact role of the PFC in fear-related behavior is still open to debate” but do not offer any evidence to support this claim. I'm not questioning the point, but only asking for citations that would make the statement credible

We agree that this problem was not clearly exposed in the manuscript, despite being of key importance. We have extended this paragraph to elaborate on the different positions and added the relevant citations (lines 116-129):

Moreover, the exact role of the PFC in fear-related behaviour is still open to debate. Original studies showed that the PFC was involved in the extinction of fear behaviour [1]. More recently, this function has been restricted to the infralimbic area, whereas it has emerged that the prelimbic area plays a role in freezing expression ([2,3], see [4] for review). Whether there is a common framework for understanding the two functions and how they are to be reconciled with PFC function in general is still unresolved [3,5,6]. In particular, the PFC is generally viewed as an integrative region providing top-down inhibitory control [7]. Fear extinction is a context-dependent top-down regulation of a previously learned behavioural response and therefore fits very well with this view of PFC function. However, a direct role for the prelimbic PFC in piloting freezing behaviour, whose main effector regions are subcortical, appears hard to reconcile with the broader role of the PFC. Since the 4Hz rhythm is important to PFC function during freezing, we use manipulation of the origin of this rhythm, the OB, to probe the function of the prelimbic area. This approach is similar to the manipulation of the theta oscillation that originates outside the hippocampus in order to understand hippocampal function.

3. Throughout the word “fear” is used in an uncritical way, ignoring a major debate in the field about the use of mental state terms like fear in relation to behaviours such as those studied. The debate, which was published in Nature Reviews Neuroscience, has received considerable attention and should be acknowledged. This is not a suggestion to about how the authors should use the term fear, but instead a suggestion that the controversy/discussion be recognized rather than ignored.

We thank the reviewer for pointing out this debate-style article that gives a very stimulating overview of how different visions of ‘fear’ are related. He/She is right to point out that setting our study in the broader context of human emotions requires questioning the extent to which freezing behaviour can be related to human fear. This point is all the more important when relating our findings to the James-Cannon debate that, since the two protagonists of the debate were interested in the conscious experience and physiology of fear respectively (point 5).

We have now clearly raised the question of how our findings relate to ‘fear’ used either as a restrictive term for human subjective experience or, more broadly as the global neural-behavioural system that includes the associated subjective experience (lines 633-647):

The above proposed interpretation attempts to reconcile behavioural and physiological data from non-human animals and reports of conscious experiences from humans. However, there remains disagreement on how to link these two approaches, in particular in the case of fear for which there is a huge controversy about whether to apply this term to non-human animals where the subjective experience of fear cannot be evaluated[8]. In fact, this issue also contributes to making the original terms of the James-Cannon debate somewhat ambiguous. James’s proposal was specifically concerned with the conscious experience of emotion whereas Cannon was studying the outward manifestations of emotion in non-human animals [9]. Although it has been argued that ‘fear’ should be reserved for the human experience, here we use the term to broadly refer to both. Indeed, neural activity and pharmacological responses that correlate with emotional behaviour in humans and non-human animals also correlate well with reports of subjective emotional experience. This suggests that effects observed in the behaviour of non-human animals can be speculated to be linked to effects in human subjective experience [8,10,11]. We can therefore tentatively suppose that if our findings are relevant for emotional behaviour in animals, there may be some parallels in both behaviour and subjective experience in humans.

4. Towards the end, it is stated: “This model clarifies the role of the PFC in fear related behaviour” and especially its role in behavioral “perseverance.” Please search PubMed to see studies that have explored the role of infralimbic and prelimbic cortex in “emotional perseveration” in the context of Pavloivan extinction.

We thank the reviewer for pointing out these references that are very relevant to our findings and that we had not fully taken into account. We have incorporated those that seemed most relevant in the discussion section. This also prompted us to give a rounder view of the relation the PFC’s function in emotions and other tasks (lines 540-557)

Although the function of the PFC remains much debated, strong evidence supports its role in inhibition both at the cognitive and motor level [12–14]. As a concrete example, in the Wisconsin Card sorting task, patients with frontal lesions asked to sort cards according to one rule (colour for example) perform like controls. However, when the rule changes to sort by shape, they fail to inhibit the old strategy and to switch to the new strategy. It has been suggested that the multiple forms of top-down regulation provided by the PFC reflect a single common process for behavioural flexibility which may extend to the domain of emotions as well [14,15]. In other words, the maintenance of an ongoing state related to decision making could be controlled by similar mechanism within the PFC than the maintenance of a fear state.

In rodents, there has been clear demonstration that the infralimbic cortex is essential for fear extinction, the reduction of fear-response to a previously threatening stimulus after its repeated non-reinforced presentation [1,16,17]. Since extinction relies on a top-down inhibition of freezing behaviour and is flexible and context-dependent, the role of the infralimbic PFC fits with the global function of the PFC as a top-down regulator of behaviour. The initial studies referred to the syndrome induced by infralimbic PFC lesions as emotional perseveration [1,17], stressing the commonality with what can be termed in parallel ‘paradigmatic perseveration’ in reference to the similar lack of behaviour flexibility in cognitive tasks [15].

5. Finally: “Our proposed model supports the James-Lange theory” because “emotional breathing entrains oscillations in the brain which in turn play an active role in regulating emotional expression.” The James-Lange Theory is not about the regulation of emotional expression but about the nature of conscious emotional feelings like fear. The last sentence comes around to conscious emotion, and implies that the results reported might also explain this. But to bring conscious emotional experience up requires a critical discussion of the emotional experience literature, and also brings us back to the issue about what fear is mentioned above.

This is indeed a key point that must be discussed in order to appraise the relevance of our findings to the James-Lange theory, since both James and Lange were trying to understand the subjective ‘emotionality’ within an emotion. We now point out in the discussion section that our findings are only relevant to this debate if we can hypothesize a link between emotional behaviours in non-human animals and subjective experience of emotions in humans. This brings us back to point 3 concerning the definition of the word ‘fear’. We therefore discussed these two issues in the paragraph quoted above.

The paper has the potential to have an important impact, and it's conceptual content should be as rigorous as the empirical work.

I would be in favor of publication if these points are addressed.

Reviewer #2 (Remarks to the Author):

In this study Bagur et al. describe a role for olfactory bulb (OB) 4 Hz oscillation in maintenance of cue-induced freezing behavior in mice. They also describe a functional connection between the OB and the prefrontal cortex (PFC), which has a preferential frequency of 4 Hz. While providing very interesting results, the study lacks important controls and the results are over interpreted.

1. First the authors do not provide evidence for a causal role of the 4Hz OB oscillation in cue-induced-freezing. To show causality of OB 4 Hz oscillation, the authors should apply a 4 Hz optogenetic stimulation during the CS-. If this stimulation increases freezing, it would demonstrate the sufficiency of OB 4 Hz oscillation to increase freezing. Indeed, by inducing a 13Hz stimulation they decrease freezing, which could be through alteration of the 4Hz oscillation, but also through alteration of AP distribution and information flow to the downstream region independently of the 4 Hz oscillation.

Similarly, the role of the 4Hz OB oscillation in maintenance rather than initiation of cue-induced freezing, need further demonstration. For example if the authors induce a 4 Hz oscillation in the OB using optogenetics after initiation of a cue-induced freezing bout, the authors should see an increase of freezing bouts length.

We thank the reviewer for his/her comments. They have led us to perform new experiments and analysis to test our proposed conceptual framework and also to substantially clarify the terms in which it is expressed. As we describe below, we have performed the experiment asked by the reviewer and we discuss how the results fit with our model. Moreover, we provide, new analysis of neural activity in the PFC in order to provide further support for the role of the 4Hz OB oscillation in maintenance.

First of all, we agree with the reviewer that the data presented in the manuscript clearly supports the necessity of the 4Hz to the freezing behaviour but does not demonstrate its sufficiency. However, it is important to note that sufficiency was not a crucial element in our hypothesis. Our behavioural analysis was in fact motivated by what seems to be the surprising consequence of the results showing that 4Hz in PFC can induce freezing and that this 4Hz rhythm originates in breathing: taken to the extreme, one could imagine that making a mouse breathing at 4Hz would therefore be sufficient to induce fear which seems difficult to believe.

Accordingly, the model we propose for the role of the 4Hz oscillation and the PFC in freezing does not imply that this oscillation is sufficient to induce freezing. The model instead proposes that once a freezing bout has been initiated - a process outside the control of the 4Hz oscillation - then the 4Hz is necessary to sustain this freezing. We therefore predict that strong 4Hz during freezing (either naturally occurring or due to stimulation) will be associated with long freezing episodes but not that imposing 4Hz outside of freezing can lead to an increase in freezing.

In fact, understanding the exact mechanism by which the OB 4Hz and PFC activity regulated freezing was a key issue in our study, and in particular because it seemed unlikely to us that they would be sufficient to drive freezing behavior. We now lay this problem out in the discussion (lines 510-533):

Taken at face value, the above results might seem to suggest that if a mouse begins to breathe at 4Hz, this will entrain the OB and in turn the PFC and therefore induce freezing. This would be surprising and seems highly unlikely, in particular since 4Hz can be observed, albeit less often, outside of freezing. In a similar vein, the role of the PFC in top-down inhibitory control seems hard to reconcile with any role in direct control fear expression. A key question is therefore how we can reconcile, both for the breathing rhythm and the PFC, the fact that they can indeed modify freezing expression with their broader functional roles which make it unlikely that they directly pilot freezing behavior.

The finding that perturbing the OB 4Hz rhythm specifically reduced maintenance of ongoing freezing episodes, independent of any effect on freezing initiation provides insight into this issue. It leads us to propose a causal chain of emotional generation that involves two major steps. First, behavioural (freezing) and physiological (breathing) responses to the threatening situation are initiated via the classical pathways of the amygdala and periaqueductal gray. Indeed, stimulation of these areas can robustly initiate freezing behaviour [18–20] and amygdala ablation modifies bout initiation [21]. Moreover, these structures are known to modify breathing, possibly via the nucleus retroambiguus [22,23]. Once initiated, the second part of the process is mediated by regular breathing. This rhythm entrains the OB then the PFC, allowing to organize neural activity in this region. This likely participates in the neural computations linked to the sustained and offset period of freezing. This is responsible for maintaining long freezing episodes (Fig6). Therefore, this second branch of the brain-body-brain loop which relies on somatic feedback (breathing) specifically regulates freezing maintenance. Based on this model, neither the PFC nor the OB 4Hz rhythm directly control freezing expression but both are necessary in order to sustain long freezing episodes.

Optogenetic stimulation at 4Hz

Aside from building on the reviewer's remark to improve the description of our conceptual framework, we also did the experiment suggested by the reviewer. We first applied 4Hz optogenetic stimulation during the CS-, as suggested by the reviewer. The freezing level remained unchanged in the presence or absence of 4Hz laser stimulation (laser ON 15,2%, laser OFF 24,3%, Wilcoxon signed rank test: Signed Rank statistic=65, p=0.773, n= 8 Chr2 mice). We also applied 4Hz optogenetic stimulation in unconditioned mice (FigR1A,B). This did not affect the freezing level, that remained low for both Chr2 and GFP mice. These results are consistent with a role for 4Hz in the maintenance of already initiated freezing episodes which are unlikely in unconditioned animals or during the CS-. In these cases, we would only expect an effect if we manipulated a brain mechanism responsible for initiating freezing.

We then also applied 4Hz optogenetic stimulation during CS+ in conditioned animals (FigR1C,D). In this case, freezing bouts have already been initiated and therefore manipulation of the OB oscillation can impact their maintenance. Strikingly, we found that optogenetic stimulation reduces freezing behaviour in Chr2 mice and not the GFP controls.

Fig R1. Optogenetic stimulation of the OB at 4Hz

A. B. Median freezing levels of unconditioned ChR2-expressing ($n=10$) and GFP-expressing controls mice ($n=5$). (Wilcoxon signed rank test: Signed Rank statistic=94, 26, $p=0.396$, 0.825, $n=5,5$; Effect size for ChR2 with/without stimulation: 0.12; Effect size ChR2 vs GFP during stimulation: 0.80)

C. D. Median freezing levels of ChR2-expressing ($n=10$) and GFP-expressing controls mice ($n=10$) during the test session. (Wilcoxon signed rank test: Signed Rank statistic=142, 103, $p=0.006$, 0.910, $n=5,5$; Effect size for ChR2 with/without stimulation: 1.65; Effect size ChR2 vs GFP during stimulation: 0.88)

This result is very similar to our original finding that 13Hz reduced the freezing levels, we therefore propose that the 4Hz stimulation is in fact having the same physiological effect: scrambling OB output and perturbing its effect in the PFC. Indeed, when we stimulate at 4Hz using optogenetics, the OB is also oscillating at around 4Hz. Thus, an externally applied stimulation at 4Hz will sometimes occur in phase and sometimes out-of-phase with the endogenous oscillation. The situation is further complicated by the fact that the actual frequency is not exactly 4Hz but varies between mice and across time. This is unavoidable in an open-loop system for which the exact phase relation between stimulation and endogenous signals are random. FigR2 shows using both simulation and measurements from our data that the phase of an open-loop stimulation relative to the endogenous oscillation rapidly converges to random (50% of stimulations in phase). This is true for both 4Hz and 13Hz, whatever the initial stimulation phase. This random stimulation will therefore perturb the oscillation both at the peak and the trough of firing, abolishing the regular, oscillatory character of the OB output.

As a result, open-loop stimulation of the OB at 4Hz, or any other frequency, will always lead to a perturbation of endogenous activity. This explains the drop in freezing levels we see in both cases. This approach is therefore not suited to demonstrate an increase in freezing by increasing the 4Hz oscillation.

However, the reviewer was indeed correct in pointing out that our model predicts that an increase in 4Hz during freezing (once it has been initiated) should lead to maintained freezing bouts. Given the large size of the OB, imposing an oscillation that would override the endogenous 4Hz oscillation would be technically very difficult to achieve. Therefore, the only way to increase the power of the 4Hz oscillation or prolong an existing oscillation as the reviewer asked would be a closed-loop system where the endogenous oscillation is detected online during freezing and a stimulation is triggered at a given phase of the oscillation. This would require extensive technical development since it necessitates a brain-machine interface which calculates the phase of the signal online and modulates the optogenetic stimulation accordingly. Moreover, since the stimulation is continuous, once initiated it will interfere with estimation of the endogenous signal, leading to non-trivial problems of online signal processing. We therefore believe that it is beyond the scope of this study and was difficult to achieve in the current sanitary situation that places limits on experimental work.

FigR2. Open-loop stimulation at any frequency results in random perturbation of endogenous 4Hz

A. Cartoon example of a 4Hz endogenous oscillation with neuron's firing at trough of stimulation (red). Delivering a stimulation (blue) at exactly the right frequency but with a random starting time - which is unavoidable using an open-loop system will lead for some episodes to in-phase stimulation and for other episodes to out-of-phase stimulation. On average the stimulation will therefore be given at a random phase and therefore neither block nor reinforce the oscillation but tend to scramble it.

B. Simulation of a 13Hz stimulation relative to a 4Hz with two different starting phases.

C. For these two starting phases, the percent of stimulation in phase quickly converges to 0.

D. The percent of stim in phase for multiple simulations starting at different phases (gray, black: mean +/- std). Red dots show measured % of stimulation in phase from individual freezing episodes during 13Hz stimulation. Note that simulations and data agree well.

E. F. G Same as B.C.D but for 4Hz stimulation. The same rapid convergence of average phase to random is seen, although the convergence is a little slower with thigh low frequency stimulation.

Previous results demonstrating the sufficiency of 4Hz

It should be pointed out that closed loop stimulation of 4Hz in the PFC during freezing has been performed by Dejean et al. 2016 and their results strongly support our own. By stimulating the PV interneurons during the ascending phase of the endogenous oscillation, they decreased 4Hz and freezing. Stimulating during the descending phase, they increased 4Hz and freezing. This strongly suggests that the in-phase stimulation designed to entrain the 4Hz oscillation is sufficient to prolong episodes.

Interestingly, although in this paper the results were not discussed in terms of maintenance and initiation, our findings allow to reinterpret them in this light. Indeed, during these experiments, mice had already initiated a freezing episode and the in phase and out of phase manipulations allowed to prematurely terminate or sustain the ongoing freezing period. Therefore, this manipulation allowed to measure the impact of the PFC 4Hz on freezing maintenance / termination. Stimulation during descending in fact maintains freezing level at a plateau close to 100% (Fig 4,f, reproduced below). Therefore, phase-specific manipulation of PFC 4Hz strongly promotes freezing maintenance.

Although these results do not directly address the role of the OB 4Hz, since we have clearly shown the origin of the PFC 4Hz is the OB, they do strongly argue for a causal role of the 4Hz oscillation in maintaining freezing episodes and for its involvement in freezing maintenance more specifically.

FigR3. Closed loop stimulation

Reproduced from Dejean et al. 2016

d, Schematic of the strategy used to inhibit dmPFC PNs during the ascending or descending phase of 4 Hz oscillation. Stim., stimulation. e, Top: representative example of freezing (black line) and kinetics of light pulse stimulations (blue ticks) during and after CS⁺ presentations 48 h (e) after conditioning. Bottom: corresponding LFP spectrogram in the 2–16 Hz band (binary logarithmic scale) during light-mediated inhibition of dmPFC assemblies in the descending phase of the oscillation. f, Left: time-resolved changes in freezing during light-mediated inhibition of dmPFC assemblies in the descending phase of the 4 Hz oscillation in PV-IRES-Cre mice infected within the dmPFC with Chr2.

Complementary analyses reinforcing the link between 4Hz – PFC activity and freezing maintenance

- ***Stronger 4Hz power is linked to more sustained and stable freezing episodes***

A key question raised by the reviewer's comment is whether 4Hz is just necessary for freezing maintenance or whether it is also sufficient to drive longer freezing bouts. Another approach to causal manipulation is to ask whether stronger 4Hz occurring spontaneously is linked to long freezing episodes (i.e. maintenance).

We have addressed this possibility with new results shown in the new figure 4 of the manuscript. There, we show that 4Hz is stronger during long (>10s) episodes of freezing. Moreover, strong 4Hz is associated with a reduction in micromovements of the head which are induced by the brief pips of the CS⁺ during freezing. This suggests that a strong 4Hz is associated with a more robust freezing which is therefore more likely to be sustained. These results reinforce the argument that 4Hz is involved in maintaining freezing episodes.

- ***Link between the firing pattern of PFC units, the OB 4Hz oscillation and freezing***

Another key point raised by the reviewer is the question of the exact link between the OB 4Hz oscillation and freezing maintenance instead of other possibilities that he/she proposes: "alteration of AP distribution and information flow to the downstream region independently of the 4 Hz oscillation". We have performed extensive analysis of PFC neural firing patterns to better understand the exact link between the 4Hz input to this region and freezing. The results are shown in figure 5 of the revised manuscript.

In brief, we show that PFC neurons show a strong and sustained difference in firing during freezing as well as a transient response at the start and end of a freezing episode – with a much stronger response at the offset of freezing than at the onset. This profile of response is suggestive of a specific control of freezing maintenance / termination. These responses are strongest in neurons that are modulated by the 4Hz oscillation and their preferred firing phase is linked to whether the neurons are activated or inhibited by freezing. Moreover, these two responses are reduced in bulbectomized animals, whereas the smaller onset response is not affected.

Altogether, this explicitly links the entrainment of neurons by the 4Hz oscillation and their encoding of freezing activity, suggesting that it is indeed the rhythmic input of the 4Hz from the OB that plays an important role in organizing PFC freezing-related activity. It moreover allows us to discuss in more detail the difference between freezing maintenance and termination and possible mechanisms that could be associated with one or the other.

Summary

We have tried to address the reviewers concerns about the causal link between 4Hz and freezing maintenance: we have performed the 4Hz stimulation experiment requested and have added complementary analyses. In summary:

- 4Hz stimulation does not induce an increase in freezing, because in an open-loop stimulation setup, the phase of stimulation is random and so has a perturbative effect (FigR1, R2). The necessary experiment of closed-loop 4Hz stimulation would require a highly technical experiment that we believe is beyond the scope of the manuscript.
- In order to more clearly link 4Hz to freezing maintenance, we provide new data showing that stronger 4Hz is associated with longer freezing bouts and freezing that is more robust to external stimulation (Fig4).
- In order to specifically link the oscillatory input to the PFC provided by 4Hz to freezing maintenance we analysed in more detail PFC neural activity. We show that the strength and phase of modulation of individual neurons by 4Hz is linked to their sustained and offset responses to freezing episodes. This argues for a specific temporal role of the 4Hz oscillation and not just the OB's overall spiking output (Fig5)

Overall, we hope these results reinforce the link between the OB 4Hz oscillation's impact on the PFC and freezing maintenance / termination. The revised manuscript therefore presents a much more rounded description of the relation of 4Hz, PFC activity and freezing and leads to the possible discussion of neural mechanisms. We thank the reviewer for having spurred these new experiments and analyses.

2. Second, the authors argue that the OB and PFC are connected, but do not even hypothesize about the pathway potentially linking these two brain regions. If the author claim a direct connection, as presented in Figure 5, they need to provide anatomical evidence for this monosynaptic connection using optogenetically assisted circuit mapping, anterograde/retrograde tracing, or electron microscopy. If present, the role of this potential direct connection should be tested using direct manipulation of the OB-PFC neurons.

The reviewer is right to point out that the question of the connection between the PFC and the OB was not discussed in our manuscript, we now discuss this still-open question in the discussion (lines 460-466) and we have modified figure 5 to reflect that the OB-PFC connection is not monosynaptic.

Overall, there is currently no evidence for a direct monosynaptic connection between the two structures. We found no report of such a link in the literature. In order to verify, we injected retrobeads in the prelimbic area and found no retrogradely labelled neurons in the OB. However, similar to the results shown in Moberly et al. we found some sparse staining the Taenia tecta which itself received input from the OB (FigR4). This region is therefore a candidate relay.

We also tested whether the piriform cortex could provide a relay by infecting piriform cortex neurons with Chr2-GFP. Both anatomy and the measurement of light-evoked responses suggest that this region does not project to the prelimbic PFC but to the infralimbic, which in turn could transmit the signal to the prelimbic. More generally, given that the respiratory rhythm has been

recorded in multiple cortical areas, the PFC may inherit it indirectly from intra-cortical connections.

These results suggest that convergent, multi-synaptic pathways provide the link between the OB and the prelimbic PFC. Whatever the pathway may be, our finding that around 45% of units were modulated, shows that it is sufficient to provide a strong entrainment of PFC activity. Indeed this value is a little larger than that previously reported for the modulation of the PFC by the theta rhythm (39%[24], 28%-46%[25]). However, this very redundant projection profile means that it is not possible to target a specific projection pathway in order to selectively abolish the 4Hz input to the PFC.

Fig-R4. Possible intermediate structures between the OB and the PFC

A. Injection site of retrobeads in the PL.

B. Cell bodies labelled by retrobeads are found in the IL and, more importantly, the DP and TTd, two regions receiving direction input form the OB.

C. ChR2-GFP unilaterally injected into the piriform cortex reveals dense fiber projections to the ventral part of the PFC (the IL) and neighbouring olfactory areas such as the TTd and DP (visible on other slices).

D. E. Stimulation of PiCx (30 x 50ms pulse) using ChR2 during acute recordings of the PFC at successive depths, given on the left-hand axis, during anesthesia shows an increase of response in both triggered LFP (B) and summed single unit activity (C) with depth. No clear response is found in the PL. Responses are z-scored.

3. Moreover the extreme vicinity between the two regions is a major confound regarding their conclusions and as they admit in the discussion, volume conduction is very likely partially responsible for the 4Hz OB PFC coherence. The proportion of the volume conduction contribution and the proportion of the of circuit connectivity contribution to the PFC 4Hz oscillation need to be discussed more carefully.

The reviewer raises a very important point: the issue of volume conduction is a crucial one in small brains such as the mouse, particularly when dealing with such large oscillations as the OB respiratory rhythm. We need to separate very clearly between our observation that PFC neurons are entrained by the 4Hz rhythm, which is the most important for our conclusions, and the origin of the LFP recorded in the PFC.

First, our results clearly support that breathing feeds back via the OB and temporally modulates PFC spiking activity. This important point is independent of the amount of PFC LFP that is due to volume conduction and based on three main arguments:

- 45% of single units in the PFC are modulated by the 4Hz activity. Therefore, even if the LFP is partly explained by volume conduction, the 4Hz oscillation arrives in the PFC and impacts firing. (Fig2I-L)
- Optogenetic driving of the OB is sufficient to entrain PFC spiking activity, showing that OB activity is sufficient to control PFC firing. (FigS6)
- Destruction of the olfactory epithelium with methimazole, although a problematic approach, clearly shows a reduction in phase locking to 4Hz in the PFC. This shows that the OB is necessary for PFC single unit locking to 4Hz. (Fig S5)

We have clarified these points in the results section (lines 210-215).

This being said, the question of the origin of the LFP recorded in the PFC is an important one. Given the proximity of the OB and the very large fields it generates, an extreme view could be that the recorded LFP are 100% due to volume conduction from the OB. We now therefore explicitly address this in the results (lines 195-207) and discussion section (lines 450-466), as well as a new supplementary figure (FigS2).

First, volume conduction cannot reflect the totality of the recorded signal because during artificial optogenetic stimulation, the frequency profile of response in the PFC is very different to the OB response: 10Hz is reduced compared to the neighbouring frequencies 7Hz and 13Hz (FigS3). The passive electrical properties of biological tissues cannot produce the non-linear transfer function between OB and PFC unveiled by our optogenetic stimulation of the OB.

In order to fully address this question, we also linked the degree of PFC-OB coherence with the depth of recording in the PFC [26]. Beforehand, we assessed PFC depth by using the average delta-wave response that is known to show a clear inversion between the deep and superficial layers (FigS2A). With 3 mice with a clear inversion, this allowed us to establish that the OB power coherence changes with depth, with the strongest coherence being found in deep PFC layers (FigS2B).

Moreover, we used bipolar derivation of deep and superficial layers to generate a pseudo-local signal, is a classical method that often eliminates volume conduction. We found that this pseudo-local signal still shows a strong coherence of PFC with the OB (FigS2C) suggesting that there is indeed a local signal that is coherent with the OB. On the contrary, bipolar derivation completely abolishes the coherence of the PFC with the HPC theta (FigS2D). The theta signal in the PFC has indeed been argued to be pure volume conduction [27], thus demonstrating that this method is efficient for identifying many volume conducted signals. Finally, although the bipolar approach does not allow to clearly identify the dipole underlying the 4Hz in the PFC, our findings of

strongest OB coherence in deep layers, is coherent with a recent CSD analysis that suggests the inversion can be found in layer V [28].

Overall, we conclude that the LFP is at least in part a genuine reflection of local processing that is linked to the specific anatomy of the PFC, even if volume conduction likely contributes. Moreover, the modulation of neurons by the 4Hz clearly indicates that the rhythm is impacting PFC local processing.

4. Third, in the result section the authors interestingly introduce and discuss previous studies that performed pharmacological inhibition or OBX. However, this section of the manuscript is not structure and these seminal studies need to be cited in the introduction of the manuscript, the results they obtain to reproduced the findings, should be described in the results section (as they are currently) and the discussion of their data in light of the previous studies should be in moved to the discussion section.

In the introduction we now explicitly discuss the results from these previous studies, since the current discrepancies in the literature provide a strong motivation for our study. We have also rewritten this section of the results (lines 262-281) to restrict it to the specific results that we provide, reserving a broader comparison of our findings with those from [29] for the discussion section (lines 473-500).

5. Overall, the study needs to be completed to support the claim of the authors, and/or the interpretation of the data need to strongly be tempered.

We hope that the reviewer finds, as we do, that the results we have added in responses to his/her questions make our conclusions more strongly supported. Moreover, reviewer 3 also asked for a more cautious rewriting of the manuscript, focusing on the main results and less on their interpretation. We have rewritten and restructured the manuscript, in particular reserving the discussion of brain-body interactions to the last section of the discussion.

6. ~ Minor Comments ~

The manuscript contains many typos and would benefit to be read by a native English speaker.

Thank you for carefully picking up these mistakes, we have made the corrections and a close rereading to avoid typos.

~ Abstract

Second sentence: add a coma after 'Here we report that in mice'.

The abstract has been modified.

The last sentence of the abstract is not understandable without reading the manuscript.

This abstract has been rewritten and we have adjusted the last sentence of the abstract to simply state the link with the James-Cannon debate, leaving elaboration for the full text:

These results provide a mechanism for the much discussed role of the bodily feedback in emotions and therefore shed light on the historical James-Cannon debate.

~ Results

Entire section: When the authors refer to cue-induced freezing, it would be helpful to specify it.

All the results in the manuscript concern cue-induced freezing, we have made this more explicit in the abstract and introduction.

First paragraph: please rephrase the description of the CS-, currently described as an unconditional stimulus. While this is technically true, this is confusing as the US is usually referring to the electric shock.

This was indeed a bad choice of wording, thank you for having pointed this out. We modified the sentence accordingly:

*In test sessions 24 hours later, animals demonstrated robust freezing in response to presentation of the **paired** stimulus (CS+) but not the **unpaired** stimulus (CS).*

Third paragraph: Replace ‘at time of testing which.’ with ‘at time of testing.’

We modified the text accordingly.

Fourth paragraph 13 HZ = 13 Hz

Sixth paragraph: replace ‘we exactly reproduced of the protocol’ with ‘we reproduced the protocol’

We modified the text accordingly.

~ Figures

- The first figure cited in the manuscript is the Figure 5. Please either move this Figure as Figure 1 or change the text.

We removed the reference to figure 5 from this sentence

- Figure 1B: please add an amplitude scale bar for the oscillation in OB and PFC. An inset with only 3 cycles enlarged would be helpful to visualize the oscillatory dynamics.

We enlarged the example figure to facilitate visualization of the oscillatory dynamics.

- Figure 1C: what is counted ? cycles ? also, please add a legend of the colors on the figure and in the figure legend

Indeed, the frequency of breathing is taken for each respiration. We modified the y axis, which is “Proportion of breaths”. We indicated the legend of the colors on the figure

- Figure 1 IJ: add y axis units = Cue-induced freezing (% Time)

We modified the figure accordingly (now Figure 3A,B).

- Figure S1I: add y axis units = Context-induced freezing (% Time), and CHR2 = ChR2

We have modified the figure to indicate 'Spontaneous freezing', since these results come from the habituation session prior to conditioning.

~ Materials and methods

~ Electrode implantation and virus injection – second paragraph

- Replace 'to obtain optimal to the glomerular layer' with 'to obtain optimal expression in the glomerular layer'

- Same sentence: remove 'to' in 'whereas to the granular'

~ Fear conditioning: last sentence replace 'they different' with 'they differed'

~ Optogenetic stimulation

- 3 shorts bouts = 3 short bouts

We have modified the text accordingly.

- What is the diameter of optic fiber used ? 300 um ?

We now specify this in the corresponding section : "3 weeks after injection, mice were bilaterally implanted with home-made optic fibres (200 μ m-diameter)[30]."

~ Discussion

- In the second paragraph of the discussion, the authors write '10Hz oscillation we evoke in the OB activity that does not lead to PFC LFP modulation whereas neighboring frequencies of 7Hz and 10Hz do elicit a response' : the authors probably mean 'frequencies of 7Hz and 13Hz do elicit a response' ?

Also, the authors should specify the Figure panel they refer to at the end of this sentence.

In line with our simplification of the main argument of the manuscript in response to reviewer 3, this has been removed.

- Please correct the use of the verb 'to argue': Individuals argue, data do not argue, but suggests
We modified the text accordingly.

- Do not use double superlative such as 'more strongly' (stronger is the adapted term)

We modified the text accordingly.

Reviewer #3 (Remarks to the Author):

Bagur et al. investigates the mechanistic underpinnings of the 4Hz oscillation present during freezing in mice. They find that respiration has an effect on freezing maintenance through an OB->PFC circuit. The authors conclude that this phenomenon represents a brain-body-brain loop.

Bagur et al. found that both bullectomy and optogenetic perturbation of OB activity reduced freezing. This is at odds with findings of Moberly et al., who found that TTX and methimazole in the OB caused prolonged freezing. Bagur et al. compared the different approaches and concluded that TTX and methimazole may have more non-specific, off-target effects. They also present correlative evidence that the respiratory rhythm is linked to termination of freezing more than initiation, and present a simple Markov model consistent with the idea that the key factor influenced by breathing is freezing maintenance.

Next, the authors went on to show that OB influences freezing probably through PFC. This was based on coherence of oscillations, resonance properties probed by optogenetic stimulation and phase locking analysis of PFC units.

1. The authors made a thorough comparison between different modes of perturbing the OB, all having different limitations. I would be inclined to believe that the techniques presented here provide a more reliable and potentially more interesting, detailed picture on how the OB oscillation influences freezing bouts specifically. At the same time, I feel this bit of information, although interesting to experts of freezing behavior in mice, is somewhat incremental and, especially, a bit detached from the sweeping conclusions the authors tend to make in the manuscript.

We thank the reviewer for his/her comments. They have led us to undertake a thorough rewriting of the manuscript as suggested by his/her later comment that we think streamlines the presentation of our results and also allows us to make clearer the relevance of our findings. Accordingly, we have added new panels and one entirely new figure and we have removed the result concerning frequency-specific communication between the PFC and OB which distracted from the main focus of the paper. We have also made a clearer separation between our results and our discussion of their interpretation in a wider context.

We now focus on clearly laying out the evidence for our main conclusion, that is that 4Hz from the OB impacts PFC firing and this contributes to regulating freezing maintenance. These results are not simply incremental since they resolve a problem in the existing literature regarding the role of 4Hz in the OB and the PFC for freezing (see point 2.), by presenting three novel findings:

- During freezing 4Hz originating in the OB and transferred to the PFC is necessary for high freezing levels: previous results from the field were contradictory
- Behavioral modelling demonstrates that 4Hz is specifically involved in freezing maintenance/termination and not initiation: this is a novel distinction for the field that we discuss again below
- PFC neurons entrained by 4Hz show freezing responses that suggest they are likely involved in freezing: links between PFC firing and fear conditioning (in particular the

CS) have been described but no functional interpretation of the precise profile had previously been proposed

The reviewer is right in pointing out that our manuscript will be of main interest to all the labs with expertise in freezing. However, freezing is a behavior used to study a vast number of different aspects of systems neuroscience and so this expertise is present in a wide range of labs with different focuses. Overall change in % of time freezing is a widely used indicator of the rules of pavlovian learning [31], extinction [32], consolidation or artificial memory reinstatement [33] and as a model of PTSD [34] to quote a few examples. Therefore, showing that freezing initiation and maintenance can be separately regulated by different brain mechanisms and that simply measuring the % of time spent freezing masks these two effects may have consequences in many fields.

2. The second half of the results, concerning PFC are also interesting, however, the OB->PFC connection is not a novel finding. In this regard, my opinion is that the authors do not properly credit Moberly et al., who reached similar conclusions about this particular connection, illuminating the origins of the 4Hz oscillation that had previously been considered enigmatic.

We agree with the reviewer that the impact of the OB on the PFC had previously been shown by Biskamp etl, [35] and more specifically during freezing by Moberly et al [29]. We have now more clearly credited these original and important findings in the introduction.

We would however like to emphasize that the aim of our study was not to demonstrate the link between the OB and the PFC but to understand the functional role of this link and to provide an explanation for the lack of consistency of the existing data in the literature. Indeed, as we develop below, the results from Moberly et al. concerning the function of the OB 4Hz were in stark contrast to those found by direct manipulation of the PFC activity. Whereas Moberly et al. found that disrupting the OB increased freezing, disruption of PFC 4Hz and the PFC in general lead to reduced freezing. Therefore, although Moberly et al showed that the origin of the 4Hz rhythm was indeed the OB, the relation between the OB and PFC based on their findings was highly problematic. In the new version of the introduction, we make clear that we aimed at understanding this discrepancy in the literature (lines 101-104).

Recent results indeed suggest that the breathing rhythm is involved in emotional behaviour [36–39]. However, as we shall briefly review, these results do not yet form a coherent picture. In rodents, auditory cue fear conditioning is the most widely used paradigm to evoke a fear-like emotional state [36], most often quantified using the robust freezing behaviour. In the prefrontal cortex (PFC), a strong 4Hz oscillation has been identified during freezing [37]. This rhythm organizes neural activity within this structure and with the amygdala. Increasing this rhythm with optogenetics increases freezing behaviour whereas its perturbation reduces freezing levels [38,39]. A recent study identified the origin of this 4Hz oscillation as being a change in breathing rhythm during freezing that is transmitted via the OB to the PFC [40]. However, pharmacological disruption of the OB and therefore the 4Hz oscillation lead to an increase in cue-induced freezing behaviour, the opposite effect of that found using direct manipulation of the PFC 4Hz [37]. Current knowledge does not allow to reconcile these contradictory results, suggesting that the function and mechanism of the respiratory-related 4Hz oscillation in fear behaviour remains to be understood.

In response to this issue, we offer a functional role for OB 4Hz that is fully compatible with previous findings from the PFC. Moreover, in the revised version of the manuscript, we have added a figure that explores in more detail the role of the 4Hz oscillation in the PFC (Fig5). In brief, we show that the entrainment of PFC units by the 4Hz oscillation is strongly linked to their freezing-related activity: units that are more strongly entrained show 1/stronger sustained and offset responses to freezing bouts and 2/ their preferred phase is predictive of whether they increase or decrease activity during freezing.

Our comparison of different methods for perturbing the OB clearly reconciles manipulation of the 4Hz both in the OB and the PFC, strongly reinforcing that the two phenomena are linked. We also provide an explanation for why the results in Moberly et al. seem to be in contradiction by reproducing their experiment and adding a number of controls. This is now laid out more clearly in the discussion section (lines 473-493) :

This raises the question of why the results of Moberly et al [40] seem to be in contradiction both with our findings manipulating the OB and findings from the literature manipulating the PFC. This study showed that the suppression of OB 4Hz, via the administration of either TTX to the OB or of methimazole to destroy the olfactory epithelium, increased freezing. Concerning TTX, a non-specific effect on freezing levels can be observed in the original publication (Fig6E of [40]) since even before tone presentation the freezing level of TTX mice is four time larger than their controls. When converting the raw time count into percentage time, this value is close to post-tone freezing levels of control mice. Concerning methimazole, here we show that it leads to a broad perturbation of overall physiology and that an increase in periods of complete immobility can be observed without any conditioning protocol, indicating that what is quantified as “freezing” may simply be immobility without relation to defensive behaviour. Given that methimazole is above all an anti-thyroid drug, these global modifications in behaviour and physiology could be attributed to changes in metabolism, in particular to the reduction in body temperature. Indeed, previous reports showed that cold-stressed mice display a drastic reduction of movement [41]. In light of these results, any clear link between methimazole effects and OB-breathing uncoupling is extremely difficult to establish. Therefore, TTX and methimazole protocols both yield non-specific increases in immobility which may not be linked to fear-related freezing but to other motor behaviours. This contrasts with partial bulbectomy and optogenetics experiments presented here which outside of the conditioning protocol have no impact on motor behaviour.

Therefore, we believe that our results resolve what was previously a contradiction in the literature and provides insight into the mechanism by which the OB influence the PFC neural activity and its functional role.

3. Taken all of these together, Bagur et al. present interesting, if rather specific and somewhat descriptive new results. However, these seem to be over-interpreted without proper credit assignment. In addition, the manuscript is hard to follow, not very logically built, and could use a thorough re-writing. For instance, the main conclusion of a brain-body-brain loop is an old concept and has long infiltrated clinical practice at a wide range of medical fields. Therefore, in the present form, I am afraid that the manuscript would not really engage a broad readership. At the same time, these results, after careful re-writing, could be important to the fear learning field.

We agree that the previous version of the manuscript was somewhat difficult to read and that the main message was not always clearly laid out. As mentioned above, we have indeed restructured the manuscript into a more logical succession of results as suggested by the reviewer.

We believe that the potential readership of the manuscript goes beyond the field of fear learning. We feel that our findings will be all the more relevant because they are not merely descriptive but are based on experimental manipulations that allow us to interventionally test the role of the OB 4Hz on both neural activity and behavior. Here, we give some other communities to whom our results are relevant:

- **Respiratory related rhythm community:** this rhythm has been observed in multiple brain areas by a growing number of groups but its functional role remained to be clearly proven. Moreover, one of the few papers testing its functional role (Moberly et al.) obtained results difficult to reconcile with the existing literature. Indeed, they showed that perturbing 4Hz in the OB increases freezing, an opposite effect to perturbing 4Hz in the PFC. Our results explain why this contradiction was only apparent and provide clear evidence that the respiratory rhythm, relayed by the OB, plays a role in one of the most intensely scrutinized behaviours in rodent research, freezing.
- **Brain rhythm community:** Many studies have described the relation of brain rhythms to behavior but it is notoriously difficult to manipulate them in order to test their functional role. In particular, manipulations often impact overall firing rate and not just the temporal organization of behaviour. Here, we show that our manipulations cannot be attributed to a change in firing rate in the PFC but still induce strong behavioural changes, arguing for a specific role in temporal organisation.
- **Affective neuroscience community:** Our results dissociate between the initiation and maintenance of freezing behaviour specifically, however this raises the question of the general interpretation of emotional behaviours and how they are regulated. This community may also be interested in the 'brain-body' link (see below).

Regarding the brain-body loop, we now only discuss this interpretation in the discussion instead of introducing it early in the manuscript. This is indeed an improvement in separating our results from their interpretation.

We think however that it is very important to evaluate our results in this context. Although this concept has indeed been around for a long time, embodied cognition is an important theme in contemporary cognitive neuroscience and its importance is a subject of interest and debate [42-44]. Our results take this debate into the domain of animal models in which mechanistic insight and more controlled manipulations are possible. This is why we have found that researchers outside the field of fear learning were interested by our results when we presented them and we have therefore left a discussion of this link in the manuscript.

4. At this point, as the presentation quality makes it hard to make a detailed assessment, I can only give examples of specific points, below.

- **Introduction, para 3. Proposing that investigating freezing initiation and termination separately 'addresses the role of PFC in fear-related behavior' is a gross overstatement.**
We have reworded this by saying that our manipulations 'probe the function of the prelimbic area' (line127)

- **There are many broken sentences, typos (e.g. 'mirroring our firing', 'regulate of freezing maintenance', 'tuned respond optimally', 'make it a promising candidate'), lab jargon and**

oversimplified statements. Without clear writing, the paper becomes extremely hard to process.

We have carefully corrected these errors and sought, through reorganization of the manuscript to respond to this criticism. We believe the main message is now much clearer.

- [1] M.A. Morgan, J.E. Ledoux, 109 (1995) 681–688.
- [2] J. Courtin, F. Chaudun, R.R. Rozeske, N. Karalis, C. Gonzalez-Campo, H. Wurtz, A. Abdi, J. Baufreton, T.C.M. Bienvenu, C. Herry, *Nature* 505 (2014) 92–96.
- [3] F. Sotres-Bayon, G.J. Quirk, *Curr. Opin. Neurobiol.* 20 (2010) 231–235.
- [4] J. Courtin, T.C.M. Bienvenu, E.Ö. Einarsson, C. Herry, *Neuroscience* 240 (2013) 219–242.
- [5] J. Courtin, T.C.M. Bienvenu, E.Ö. Einarsson, C. Herry, *Neuroscience* 240 (2013) 219–242.
- [6] F. Sotres-Bayon, C.K. Cain, J.E. LeDoux, *Biol. Psychiatry* 60 (2006) 329–336.
- [7] J.M. Fuster, *Neuron* 30 (2001) 319–333.
- [8] D. Mobbs, *Nat. Neurosci.* (n.d.).
- [9] P.C. Ellsworth, *Psychol. Rev.* 101 (1994) 222–9.
- [10] M.S. Fanselow, Z.T. Pennington, *Am. J. Psychiatry* 174 (2017) 1120–1121.
- [11] J.E. LeDoux, D.S. Pine, *Am. J. Psychiatry* 173 (2016) 1083–1093.
- [12] B. Milner, *Arch. Neurol.* 9 (1963) 90.
- [13] J.M. Birrell, V.J. Brown, *J. Neurosci.* 20 (2000) 4320–4324.
- [14] D.G. Dillon, D.A. Pizzagalli, *Appl. Prev. Psychol.* 12 (2007) 99–114.
- [15] M.D. Hauser, *Curr. Opin. Neurobiol.* 9 (1999) 214–222.
- [16] J. Courtin, (2013).
- [17] M.A. Morgan, J. Schulkin, J.E. Ledoux, *Behav. Brain Res.* 146 (2003) 121–130.
- [18] B.S. Kapp, M. Gallagher, M.D. Underwood, C.L. McNall, D. Whitehorn, *Brain Res.* 234 (1982) 251–262.
- [19] R. Bandler, J.L. Price, K.A. Keay, *Prog. Brain Res.* 122 (2000) 333–349.
- [20] E.J. Kim, O. Horovitz, B.A. Pellman, L.M. Tan, Q. Li, G. Richter-Levin, J.J. Kim, *Proc. Natl. Acad. Sci.* 110 (2013) 14795–14800.
- [21] S. Maren, W. Holt, *Behav. Brain Res.* 110 (2000) 97–108.
- [22] H.H. Subramanian, G. Holstege, *J. Comp. Neurol.* 521 (2013) 3083–3098.
- [23] H.H. Subramanian, G. Holstege, *J. Neurosci.* 29 (2009) 3824–3832.
- [24] K. Benchenane, A. Peyrache, M. Khamassi, P.L. Tierney, Y. Gioanni, F.P. Battaglia, S.I. Wiener, *Neuron* 66 (2010) 921–936.
- [25] A. Sirota, S. Montgomery, S. Fujisawa, Y. Isomura, M. Zugaro, G. Buzsáki, *Neuron* 60 (2008) 683–697.
- [26] D. Contreras, M. Steriade, *J. Neurosci.* 15 (1995) 604–622.
- [27] A. Sirota, S. Montgomery, S. Fujisawa, Y. Isomura, M. Zugaro, G. Buzsáki, *Neuron* 60 (2008) 683–697.
- [28] N. Karalis, A. Sirota, *BioRxiv* (2018) 392530.
- [29] A.H. Moberly, M. Schreck, J.P. Bhattarai, L.S. Zweifel, W. Luo, M. Ma, *Nat. Commun.* 9

- (2018).
- [30] D.R. Sparta, A.M. Stamatakis, J.L. Phillips, N. Hovelsø, R. Van Zessen, G.D. Stuber, *Nat. Protoc.* 7 (2012) 12–23.
 - [31] T.J. Madarasz, L. Diaz-Mataix, O. Akhand, E.A. Ycu, J.E. LeDoux, J.P. Johansen, *Nat. Neurosci.* 19 (2016) 965–972.
 - [32] C. Herry, S. Ciocchi, V. Senn, L. Demmou, C. Müller, A. Lüthi, *Nature* 454 (2008) 600–606.
 - [33] S. Ramirez, X. Liu, P.A. Lin, J. Suh, M. Pignatelli, R.L. Redondo, T.J. Ryan, S. Tonegawa, *Science* (80-.). 341 (2013) 387–391.
 - [34] J. Baek, S. Lee, T. Cho, S.W. Kim, M. Kim, Y. Yoon, K.K. Kim, J. Byun, S.J. Kim, J. Jeong, H.S. Shin, *Nature* 566 (2019) 339–343.
 - [35] J. Biskamp, M. Bartos, J.F. Sauer, *Sci. Rep.* 7 (2017) 1–11.
 - [36] J.E. LeDoux, *Proc. Natl. Acad. Sci.* 111 (2014) 2871–2878.
 - [37] S. Fujisawa, G. Buzsáki, *Neuron* 72 (2011) 153–165.
 - [38] C. Dejean, J. Courtin, N. Karalis, F. Chaudun, H. Wurtz, T.C.M. Bienvenu, C. Herry, *Nature* 535 (2016) 420–424.
 - [39] N. Karalis, C. Dejean, F. Chaudun, S. Khoder, R. R Rozeske, H. Wurtz, S. Bagur, K. Benchenane, A. Sirota, J. Courtin, C. Herry, *Nat. Neurosci.* 19 (2016) 605–612.
 - [40] A.H. Moberly, M. Schreck, J.P. Bhattarai, L.S. Zweifel, W. Luo, M. Ma, *Nat. Commun.* 9 (2018).
 - [41] D. Sutoo, K. Akiyama, H. Takita, *Pharmacol. Biochem. Behav.* 40 (1991) 423–428.
 - [42] H.D. Park, S. Correia, A. Ducorps, C. Tallon-Baudry, *Nat. Neurosci.* 17 (2014) 612–618.
 - [43] A.K. Seth, *Trends Cogn. Sci.* 17 (2013) 565–573.
 - [44] Chemero, Anthony, *Radical Embodied Cognitive Science*, MIT Press, 2009.

Reviewers' Comments:

Reviewer #1:

Remarks to the Author:

The authors have done a good job of addressing my concerns. The only remaining issue is with this statement:

This suggests that effects observed in the behaviour of non-human animals 645 can be speculated to be linked to effects in human subjective experience [75,77,78]. We can646 therefore tentatively suppose that if our findings are relevant for emotional behaviour in animals,647 there may be some parallels in both behaviour and subjective experience in humans.

Personally, I think this is too strong a conclusion. But I will leave it to the editors to decide whether to require any modification.

Reviewer #2:

Remarks to the Author:

The authors provide a greatly improved manuscript along with a very extensive and thorough rebuttal. They addressed all my concerns, and I only have small remaining comments:

1. Although the ideal causal experiment would have been to perform the 4Hz optogenetic stimulation in closed loop (which I agree is beyond the scope of this study) the authors should include the Rebuttal Figure Figure 1 (4Hz opto stim) in the Supplementary Figure 6. Along those line, I would advise the authors to consider sinusoidal opto-stimulations rather than pulse trains, if they perform the closed loop experiments in the future.

2. I strongly suggest the authors include a Figure of the correlation between the 4Hz power and the freezing bout duration (during CS+, CS- and no CS). This graph has the potential to further support their conclusions of the critical role of 4Hz oscillations in the maintenance rather than the initiation of freezing behavior.

In conclusion, the authors have done a very interesting and rigorous study showing the role of 4 Hz breathing in conditioned freezing behaviors in mice, and identified neural mechanisms underlying, at least in part, those breathing/freezing interactions.

Minor Comments

Please specify the duration of each optogenetic light pulse (1, 5, 10 ms?).

Results: Line 265: same does = same doses

Discussion: « results might seem to suggest » is using 3 conditional words. One is enough: replace with « results suggest »

In the same sentence, the authors say 'induce freezing'. However, considering their result, 'maintain' might be more appropriate.

Following sentence: "...any role in direct control *of* fear expression."

Figure 2B-D : the axis title are too small

Figure 2B : use colors for the inset (including the axis title : Fz in blue)

Figure 2C : please use 2 different colors for OB-PFC and PFC-OB

Figure 3C : make arrowheads smaller

Figure 5A : please add a z-score scale (dark blue = -1 ? yellow = +1 ?)

Figure S3 and S6 should be merged in one Figure.

Figure S3 : please write the 2 last lines of the legend on the same page as the Figure.

Figure S6A : please add the time scale

Reviewer #3:

Remarks to the Author:

The revised manuscript by Bagur, Lefort et al. has much improved and reads much clearer. In the previous round I wrote 'At this point, the presentation quality makes it hard to make a detailed assessment' – this is now clearly not the case, but at this point, I have a few major points, most importantly the first one below.

1. As an important aspect that still needs to be sorted out, the Authors should be clearer on anatomy. Prelimbic and infralimbic cortices have very different assumed functions in fear response regulation, as the Authors also point out, and this distinction is important to the theory presented strongly in the Discussion. Therefore, it feels insufficient to just use 'PFC' in the Results. Coordinates in the Methods indicate prelimbic, but this should be made clear, confirmed by a serious attempt of track reconstruction, demonstrated in figures.

2. Claims about the Markov model, crucial to the Authors' interpretation that the 4Hz oscillation is about maintenance, seems insufficiently presented. '...ablation or perturbation of the OB specifically reduces freezing maintenance probability (P_{Fz}/Fz), On the contrary, P_{Act}/Act ... is unaffected.' – I may be missing this information but are the fitted model parameters presented? This statement is not convincing without that information. This should be provided for both manipulations (OBX, ChR2). Why is the number of freezing bouts only presented for ChR2 in Fig.3 and not OBX, including the model prediction?

3. The Authors call the two responsive cell types 'transition active' and 'sustained active' in Fig.5. However, panels A, D and E clearly show that these cells have very similar dynamics with an opposite sign. Therefore, calling them transition active and sustained seems misleading and I suggest explicitly pointing out their opposite nature.

Minor points:

The language has improved but can use further improvement – many commas missing, there are a number of typos (line 206, 'therefore, at least'; line 222 please revise the sentence, line 291 'our results', line 848: solid line and dashed line, line 905 '13 Hz laser stimulation', line 1175 unfinished sentence).

Lines 273-277 – this is not clear to me.

References seem to be off – e.g. references to Fujisawa (25 in the ref. list) in lines 101-114.

Where was the virus injected (coordinates), and what construct exactly?

Fig.4A – Please indicate 'onset' and 'offset' in the figure.

REVIEWER COMMENTS

NB: substantial modifications to the text have been highlighted in **bold and highlighted** in the manuscript to allow easy reference by the reviewers.

Reviewer #1 (Remarks to the Author):

The authors have done a good job of addressing my concerns. The only remaining issue is with this statement:

This suggests that effects observed in the behaviour of non-human animals 645 can be speculated to be linked to effects in human subjective experience [75,77,78]. We can646 therefore tentatively suppose that if our findings are relevant for emotional behaviour in animals,647 there may be some parallels in both behaviour and subjective experience in humans.

Personally, I think this is too strong a conclusion. But I will leave it to the editors to decide whether to require any modification.

We are pleased that the reviewer finds the manuscript improved along the lines he / she suggested, we feel he / she helped us make significant conceptual improvements.

We aimed the quoted sentence to present this as a proposal (hence the 'tentatively'), but if the reviewer finds that the wording is still too strong, we propose this modification:

We therefore tentatively propose that it may be interesting to study the link between bodily feedback and the duration of emotional bouts as subjectively experienced in humans, as suggested above in reference to spinal cord patients. (L669-672)

Reviewer #2 (Remarks to the Author):

The authors provide a greatly improved manuscript along with a very extensive and thorough rebuttal. They addressed all my concerns, and I only have small remaining comments:

We are pleased that the reviewer feels his / her concerns have been addressed, they led us to include important data and analyses, in particular concerning neural activity, and so helped substantially improve the manuscript.

1. Although the ideal causal experiment would have been to perform the 4Hz optogenetic stimulation in closed loop (which I agree is beyond the scope of this study) the authors should include the Rebuttal Figure Figure 1 (4Hz opto stim) in the Supplementary Figure 6. Along those line, I would advise the authors to consider sinusoidal opto-stimulations rather than pulse trains, if they perform the closed loop experiments in the future.

We have moved the figure to include it in the supplementary materials (Figure S11) and reference it in the text (L259). In the legend of his figure, we also included a brief commentary on the interpretation of these results and the discussion of sufficiency / necessity that was raised in the process of review since this is an important point.

The reviewer is also right to point out that sinusoidal stimulations would likely improve our current experimental approach. We thank him / her for pointing this out, we will bear it in mind in future studies.

2. I strongly suggest the authors include a Figure of the correlation between the 4Hz power and the freezing bout duration (during CS+, CS- and no CS). This graph has the potential to further support their conclusions of the critical role of 4Hz oscillations in the maintenance rather than the initiation of freezing behavior.

A correlation between 4Hz power and freezing duration is indeed a prediction of our framework. We find a good correlation between 4Hz power and freezing bout duration, during all periods (CS+, CS-, no CS) (FigS14 D). Interestingly, we also found that this correlation is not significant using the 4Hz power from the first 2s after freezing onset but preserved using the last 2s before freezing offset (FigS14A,B,C). This is coherent with our observation that 4Hz is more strongly associated with freezing offset than onset and further corroborates our model in which 4Hz participates in the control of freezing offset / maintenance.

This figure lends strong support to our findings and we have added it as figure S14 references in lines 382-386. We thank the reviewer for this suggestion.

Fig S14. Correlation between the duration of a freezing epoch and the 4Hz power

A. B. C. Correlation between the duration of a freezing bout and the 4Hz power (A) at onset (during the first 2s), (B) during the full bout and (C) just before offset (during the last 2s). Inserts indicate the epochs used in reference to the evolution of 4Hz power during freezing.

D. Correlation between the duration of a freezing bout and the 4Hz power during the full bout as in B but split into three different epochs of the test session: CS-, CS+, absence of CS. For all epochs, the correlation between 4Hz and freezing duration is highly significant.

In conclusion, the authors have done a very interesting and rigorous study showing the role of 4 Hz breathing in conditioned freezing behaviors in mice, and identified neural mechanisms underlying, at least in part, those breathing/freezing interactions.

Minor Comments

Please specify the duration of each optogenetic light pulse (1, 5, 10 ms?).

The light pulse duration is not constant, we stimulated at different frequencies with a 50% duty cycle. We have added this to the Material and Methods section of the manuscript (line 1112).

Results: Line 265: same does = same doses

Corrected.

Discussion: « results might seem to suggest » is using 3 conditional words. One is enough: replace with « results suggest »

Corrected.

In the same sentence, the authors say ‘induce freezing’. However, considering their result, ‘maintain’ might be more appropriate.

We have kept the word ‘induce’ because here we wanted to insist on the idea that the simplest interpretation of our results could be that 4Hz directly induces freezing, in order to contrast this with our results that 4Hz maintains already initiated episodes but does not induce them.

Following sentence: “...any role in direct control *of* fear expression.”

Corrected.

Figure 2B-D : the axis title are too small

Figure 2B : use colors for the inset (including the axis title : Fz in blue)

Figure 2C : please use 2 different colors for OB-PFC and PFC-OB

Figure 3C : make arrowheads smaller

Figure 5A : please add a z-score scale (dark blue = -1 ? yellow = +1 ?)

Figure S3 and S6 should be merged in one Figure.

Figure S3 : please write the 2 last lines of the legend on the same page as the Figure.

Figure S6A : please add the time scale

These recommendations have been followed.

Reviewer #3 (Remarks to the Author):

The revised manuscript by Bagur, Lefort et al. has much improved and reads much clearer. In the previous round I wrote ‘At this point, the presentation quality makes it hard to make a detailed assessment’ – this is now clearly not the case, but at this point, I have a few major points, most importantly the first one below.

We are pleased that our re-writing of the manuscript has allowed the reviewer to make a detailed assessment of our results. We address his / her questions below, in particular regarding histology which indeed is a crucial control given our discussion of the respective roles of the prelimbic and infralimbic areas.

1. As an important aspect that still needs to be sorted out, the Authors should be clearer on anatomy. Prelimbic and infralimbic cortices have very different assumed functions in fear response regulation, as the Authors also point out, and this distinction is important to the theory presented strongly in the Discussion. Therefore, it feels insufficient to just use ‘PFC’ in the Results. Coordinates in the Methods indicate prelimbic, but this should be made clear, confirmed by a serious attempt of track reconstruction, demonstrated in figures.

The reviewer is correct in pointing out that the prefrontal cortex is composed of multiple subregions (the prelimbic, the infralimbic and the cingulate in the mouse) and that this regionalization plays a key part in the interpretation of our results. Indeed a number of studies indicate that whereas the ventral PFC (infralimbic) is involved in fear extinction, the dorsal regions (prelimbic and cingulate) have been proposed to be involved in fear expression itself. Most studies have focused on the opposition between prelimbic and infralimbic areas. Indeed less is clearly established regarding the cingulate cortex: whereas lesion and pharmacological inactivation provided evidence that the region controlled fear acquisition exclusively [1], more recent optogenetic studies have shown that it can directly impact freezing expression [2]. Given this ambiguity and its possible role in freezing expression, studies on PFC 4Hz prior to our own, grouped together the prelimbic and cingulate areas, referring to them as the dorsomedial PFC (dmPFC) [3,4]. Based on this and the histology results we detail below, we have now opted to refer systematically to the dmPFC in our discussion of our results. Indeed the key distinction we make is between the infralimbic control of fear extinction, which seems compatible with the broader top-down control of the PFC, and the dmPFC control of fear expression, which we argue is a priori less compatible with this role but can be reconciled using our framework of regulation of freezing maintenance.

Our interpretation therefore relies on our recordings being located within the dmPFC region and not the infralimbic region and the reviewer is right to request a thorough histological verification. We have made a systematic survey of the location of electrodes that we implanted in the PFC, paying particular attention to the location of tetrodes, in order to ascertain in which subregion they were located. These locations are provided below and in Fig S1A,B.

Fig.S1 Electrode localisation

A. The recording sites were reconstructed *post hoc* from coronal brain sections as shown in this example. The white arrow indicates the electrode track. Black bar: 500 μm

B. Recording locations for 18 mice (5 / 18 mice were implanted bilaterally) with antero-posterior distance from Bregma (mm) on the right for each section. Electrodes were localized to target areas in dorsomedial prefrontal cortex (shaded in grey). Colored dots represent recordings sites of animals in which 100% tetrodes were in prelimbic and cingulate cortex only and which were used for neural analysis whereas grey dots show excluded locations. Dots are color coded by animal.

C. Neural analysis presented in Figure 5 is performed on neurons located both in the cingulate and prelimbic cortex. In a subset of animals tetrodes were exclusively located in one of the two cortices, allowing for separate analysis (cingulate cortex on top, $n = 69$ and prelimbic cortex below, $n = 57$). Both groups of neurons show the same dynamics as the whole neuron population but the effects are clearer for prelimbic neurons.

Left. Z-scored response of units during normalized freezing episode. Error bars are SEM. Note that for all units, the response has been sign-corrected (units that increase firing during freezing are multiplied by -1) to allow averaging of responses that displays the amplitude of their freezing modulation, independent of the sign.

Middle. Averaged response (absolute value) of units at onset (yellow) and offset (purple) of freezing.

Right. Percent of significantly modulated units by OB 4Hz (black).

Overall, we found that 67% of the recording sites and 76% of the single units in our studies could be unambiguously located in either the prelimbic or the cingulate cortex. Most of the other units being recorded too laterally and dorsally in the motor cortex. Only two mice in fact had recording sites in the infralimbic cortex.

In order to analyse exclusively the activity of single units located in the dmPFC, we restricted our dataset to animals in which 100% tetrodes could be identified in this area (134 units). We have rerun the analysis on single units and updated the manuscript. Our results are unchanged, with only a slight loss in statistical power due to the smaller number of neurons (Fig5), suggesting that our findings are robust. We can therefore conclude that dmPFC single units which are entrained by OB 4Hz display stronger sustained and offset responses to freezing consistent with a role in regulating freezing maintenance via the 4Hz oscillation.

To assess the respective roles of prelimbic and cingulate regions, we used mice with tetrodes exclusively in the prelimbic or cingulate regions, we compared the responses of units from the regions. We show in FigS1C that the modulation of units by 4Hz and their stronger response to freezing offset than onset (the main results of our neuronal analysis) are found both in cingulate and prelimbic cortex, with being slightly stronger effects in the prelimbic region. This data supports an involvement of both regions composing the dmPFC in the control of freezing expression via a specific regulation of its maintenance / termination.

2. Claims about the Markov model, crucial to the Authors' interpretation that the 4Hz oscillation is about maintenance, seems insufficiently presented. '...ablation or perturbation of the OB specifically reduces freezing maintenance probability ($P_{Fz/Fz}$), On the contrary, $P_{Act/Act}$... is unaffected.' – I may be missing this information but are the fitted model parameters presented? This statement is not convincing without that information. This should be provided for both manipulations (OBX, Chr2). Why is the number of freezing bouts only presented for Chr2 in Fig.3 and not OBX, including the model prediction?

We have tried to improve the clarity of our manuscript regarding the use of the Markov model and our use of the number of freezing bouts for the model prediction (lines 321-354).

Model validation : The model has two parameters (probability of initiation ($1-P_{Act-Act}$) and maintenance $P_{Fz/Fz}$) that are not fitted to the data but in fact calculated directly from the data. Indeed, as we demonstrate in the supplementary discussion these two probabilities can be calculated from the average freezing and active bout durations. We then validate the model by simulating data based on these two parameters and measuring two variables that can then be compared to the data: percent of time spent freezing and number of freezing bouts. We have modified to FigS12A to make a flowchart (see below) in order to more clearly present the steps of this analysis.

Model validation for bulbectomy experiments : The reviewer is right to point out that in the previous version of the manuscript we did not provide the evidence of model validation for bulbectomy data. We have therefore added the bulbectomy data to FigS12C (see lines 324-327). In all cases the correlation coefficient between data and model is high (>0.8) and so the model can be used both for bulbectomy and optogenetic data.

Analysis of number of freezing bouts : In our discussion of the Markov model, we use the number of freezing bouts in order to make two different points:

- First, that the effect of perturbing 4Hz can be interpreted as a pure effect on freezing maintenance. To do this we show in Fig3E ('model prediction') that the number of bouts during the laser on period can be correctly modelled by keeping fixed the initiation probability to that of the laser off period and only recalculating the maintenance probability ($P_{Fz/Fz}$) during the laser on period. Therefore, we show that the change between laser on and laser off, and therefore the effect of 4Hz perturbation, can be fully reproduced with a change in freezing maintenance alone. This analysis can only be performed using optogenetic data since it relies on having paired data for mice before and after the intervention which is possible for optogenetics (laser off vs laser on) but not for bulbectomy.
- Second, that this parameter may be misleading when interpreted without the correct framework. We wanted to stress this methodological point by showing that the number of freezing bouts may behave counter-intuitively since it depends both on freezing initiation and maintenance. In order to illustrate this, we underline the fact that although the mice freeze less, the number of freezing bouts increases whereas one might expect the % of freezing and number of freezing bouts to increase or decrease together. We therefore propose that our simple Markov model is a good tool to allow for clear interpretation of freezing data.

We have re-written this section in order to clarify these two ideas (sufficiency of freezing maintenance and methodological warning) by devoting a separate paragraph to each (L337-346 and L348-354 respectively).

A

B

C

Figure 12. Markov model of freezing behaviour

A. Flowchart showing the establishment and validation of the Markov model used to model freezing. Based on freezing data (1), the two probabilities ($P_{Act/Act}$ and $P_{Fz/Fz}$) can be directly calculated from the average active and freezing bout durations (2, see supplementary discussion for demonstration). Based on these two probabilities, time courses of freezing activity can be stimulated : at each time step, depending on the current state (Fz or Act), the next state is randomly selected according to these probabilities (3). Finally, to establish the quality of the model, two other parameters not used to calibrate the model, the percent of time spent freezing and the number of freezing bouts, are estimated from these simulations and compared to data (4).

B.C. Correlation of data with simulation-based prediction of the Markov model for the total percent of time spent freezing and the number of freezing bouts for GFP and Chr2 animals (B) and sham and bulbectomized animals (C). Note that in all cases, there is a high correlation coefficient (all p values $< 1E-8$) and points lie close to the identity line shown in black, indicating an excellent prediction by the model.

3. The Authors call the two responsive cell types ‘transition active’ and ‘sustained active’ in Fig.5. However, panels A, D and E clearly show that these cells have very similar dynamics

with an opposite sign. Therefore, calling them transition active and sustained seems misleading and I suggest explicitly pointing out their opposite nature.

The reviewer is right to point out that these two populations show similar dynamics but with an opposite sign. The names we had initially given indeed seemed to suggest otherwise and we have renamed them “sustained-off/transition-on” and “sustained-on/transition-off” and have explicitly pointed out their opposite nature in the text (L398-404).

Minor points:

The language has improved but can use further improvement – many commas missing, there are a number of typos :

line 206, ‘therefore, at least’;

line 222 please revise the sentence,

line 291 ‘our results’,

line 848: solid line and dashed line, (now changed to black and green)

line 905 ‘13 Hz laser stimulation’,

line 1075 unfinished sentence).

We have corrected these points.

Lines 273-277 – this is not clear to me.

We have tried to clarify this passage in order to justify our use of the Moberly et al. protocol but omitting conditioning (L278-281). Based on results showing that methimazole perturbs multiple behavioural and physiological measures, we hypothesize that it could also affect overall levels of movement. This might lead to an observed change in “freezing” measured but without any specific effect of methimazole on fear behaviour or freezing, simply because of motor changes (potentially an increase in fear-unrelated immobility). This is why we reproduced the protocol without conditioning, to see if we would measure any changes in “freezing” simply due to methimazole’s effects on motor behaviour and we indeed found this to be the case.

References seem to be off – e.g. references to Fujisawa (25 in the ref. list) in lines 101-114.

The reviewer is right, something had gone wrong with the references, this has been corrected.

Where was the virus injected (coordinates), and what construct exactly?

We have now provided the full details of our protocol Material and Methods (line 1036)

A total of 17 GAD-Cre-line mice were injected with AAV5-ChR2 virus (AAV-EF1a-DIO-hChR2(H134R)-EYFP, 15 mice) and control AAV5-GFP virus (AAV-EF1a-DIO-EYFP , 7 mice) in bilateral OBs. Multiple injection sites were used to obtain optimal expression in the glomerular layer (FigS9A) since these interneurons are thought to generate the OB slow oscillations whereas the granular layer interneurons are involved in OB fast oscillations[5].

7 ChR2	ventro-medial AP +5.2, ML 0.5, DV 2.1	300 nL each
7 GFP	dorsomedial AP +5.2, ML 0.5, DV 1.2 ventro-lateral AP +5.2, ML 1.7, DV -1.7 dorsolateral AP +5.2, ML 1.7, DV -0.5	
4 ChR2	ventro-medial AP +5.2, ML 0.5, DV 2.1 dorsomedial AP +5.2, ML 0.5, DV 1.2 ventro-lateral AP +5.2, ML 1.7, DV -1.7.	200nL each
4 ChR2	AP +4.5, ML +0.8, DV -1.6	600 nL each

Fig.4A – Please indicate ‘onset’ and ‘offset’ in the figure.

We have added this.

REFERENCES

- [1] S. Bissière, N. Plachta, D. Hoyer, K.H. McAllister, H.R. Olpe, A.A. Grace, J.F. Cryan, *Biol. Psychiatry* 63 (2008) 821–831.
- [2] J. Jhang, H. Lee, M.S. Kang, H.S. Lee, H. Park, J.H. Han, *Nat. Commun.* 9 (2018) 1–16.
- [3] N. Karalis, C. Dejean, F. Chaudun, S. Khoder, R. R Rozeske, H. Wurtz, S. Bagur, K. Benchenane, A. Sirota, J. Courtin, C. Herry, *Nat. Neurosci.* 19 (2016) 605–612.
- [4] C. Dejean, J. Courtin, N. Karalis, F. Chaudun, H. Wurtz, T.C.M.M. Bienvenu, C. Herry, *Nature* 535 (2016) 420–424.
- [5] I. Fukunaga, J.T. Herb, M. Kollo, E.S. Boyden, A.T. Schaefer, *Nat. Neurosci.* 17 (2014) 1208–1216.

}

Reviewers' Comments:

Reviewer #3:

Remarks to the Author:

The Authors convincingly addressed my points. I have no further concerns.

REVIEWERS' COMMENTS

Reviewer #3 (Remarks to the Author):

The Authors convincingly addressed my points. I have no further concerns.

We are pleased the reviewer has been convinced by our answers and thank him/her for the suggestions for the new analysis and writing.